

# Investigating the effect of silicate and calcium based ocean alkalinity enhancement on diatom silicification

Aaron Ferderer[1,2], Kai G. Schulz[3], Ulf Riebesell[4], Kirralee G. Baker[1,5], Zanna Chase[1], Lennart T. Bach[1]

[1]Institute for Marine and Antarctic Studies, Ecology & Biodiversity, University of Tasmania, Hobart, TAS, Australia.
[2]National Collections and Marine Infrastructure, Commonwealth Scientific and Industrial Research Organisation, Hobart, Tasmania, Australia
[3]Faculty of Science and Engineering, Southern Cross University, Lismore, NSW, Australia
[4]Marine Biogeochemistry, Biological Oceanography, GEOMAR Helmholtz Centre for Ocean Research Kiel, Kiel, Germany
[5]The Australian Centre for Excellence in Antarctic Science, University of Tasmania, Hobart, Tasmania 7001, Australia (ACEAS)

*Correspondence to:* Aaron Ferderer (aaron.ferderer@utas.edu.au)



**Abstract.** Gigatonne-scale atmospheric carbon dioxide removal (CDR) will almost certainly be needed to supplement the emission reductions required to keep global warming between 1.5 - 2°C. Ocean alkalinity enhancement (OAE) is an emerging marine CDR method with the addition of pulverized minerals to the surface ocean being one widely considered approach. A concern of this approach is the potential for dissolution products released from minerals to impact phytoplankton communities. We conducted an experiment with 10 pelagic mesocosms (M1 – M10) in Raunefjorden, Bergen, Norway to assess the implications of simulated silicate- and calcium-based mineral OAE on a coastal plankton community. Five mesocosms (M1, M3, M5, M7 and M9) were enriched with silicate (~75 µmol $L^{-1}$ $Na_2SiO_3$), alkalinity along a gradient from 0 to ~600 µmol $kg^{-1}$, and magnesium in proportion to alkalinity additions. The other five mesocosms (M2, M4, M6, M8, M10) were enriched with alkalinity along the same gradient and calcium in proportion to alkalinity additions. The experiment explored many components of the plankton community, from microbes to fish larvae, and here we report on the influence of mineral based OAE on diatom silicification. Macronutrients (nitrate and phosphate) limited silicification at the onset of the experiment until nutrient additions on day 26. Silicification was significantly greater in the silicate-based mineral treatments, with silicate concentrations limiting silicification in the calcium-based treatment. The degree of silicification varied significantly between genera, and genera specific silicification also varied significantly between alkalinity mineral sources, with the exception of *Cylindrotheca*. *Pseudo-nitzschia* was the only genus affected by alkalinity, whereby silicification increased with increasing alkalinity during some periods of the experiment. No other genera displayed significant changes in silicification as a result of alkalinity increases between 0 and 600 µmol $kg^{-1}$ above natural levels. Nor did we observe any indication of interactive effects between simulated mineral dissolution products and changes in carbonate chemistry. Previous experiments have provided evidence of alkalinity effects on diatoms underscoring the necessity for further studies under a range of boundary/environmental conditions to extract a more robust pattern of diatom responses to OAE. In summary, our findings suggest limited genus-specific impacts of alkalinity on diatoms, while also highlighting the importance of understanding the full breadth of different OAE approaches, their risks, co-benefits, and potential for interactive effects.

## 1 Introduction

Limiting global average surface temperature rise to 1.5 - 2°C above pre-industrial levels necessitates rapid reductions in global $CO_2$ emissions as well as sustained atmospheric carbon dioxide removal (CDR) (IPCC, 2021; Van Vuuren et al., 2018). However, prior to considering the implementation of large scale CDR methods it is critical to assess the potential ecological impacts of these methods (Bach et al., 2019; Fuss et al., 2018; Renforth and Henderson, 2017).

Ocean alkalinity enhancement (OAE) is considered to be a promising marine CDR method due to its potential to remove carbon at a gigatonne scale (Burt et al., 2021; Feng et al., 2017; He and Tyka, 2023; Ilyina et al., 2013; Keller et al., 2014; Paquay and Zeebe, 2013). There are various approaches to implementing OAE, each with techno-economic and environmental advantages and disadvantages (Renforth and Henderson, 2017). Irrespective of the method, all approaches aim to increase the capacity of the ocean to store atmospheric $CO_2$ by increasing the alkalinity of seawater through the addition of substances which increase alkalinity, the removal of acid and/or neutralisation of protons in seawater, all of which will increase seawater pH (Eisaman et al., 2023). One widely



discussed method involves the addition of pulverized alkaline minerals such as magnesium silicates or calcium hydroxides to the surface ocean (Bach et al., 2019; Kheshgi, 1995; Renforth and Henderson, 2017).

In order for minerals to be suitable they must be alkaline, have relatively rapid dissolution rates, and are ideally inexpensive, readily available, and contain minimal potential contaminants (Hartmann et al., 2013; Renforth and Henderson, 2017). Minerals fitting some or most of these criteria include olivine, a silicate-based naturally
occurring mineral, as well as quick and/or hydrated lime which are anthropogenic calcium-based minerals (Renforth and Henderson, 2017). However, the effects of dissolution products derived from these minerals on marine communities is yet to be fully assessed (Bach et al., 2019). Indeed, hotspots of dissolution products from mineral-based OAE will inevitably occur at sites of mineral additions resulting in high concentrations of e.g., $Mg^+$, $Si(OH)4$, $Ca^+$, increased pH, and trace metals (Hartmann et al., 2013). Dissolution products may act to
fertilize some organisms while inhibiting others, potentially leading to shifts in plankton communities (Bach et al., 2019; Guo et al., 2022; Hutchins et al., 2023). For example, calcium-based minerals are hypothesized to benefit pelagic and benthic calcifiers, with some studies supporting this (Albright et al., 2016; Bach et al., 2015; Gore et al., 2019) while others found neutral responses (Gately et al., 2023). In contrast, the dissolution of silicate based minerals is expected to benefit silicifying plankton species including diatoms (Bach et al., 2019; Egge and Aksnes,
1992; Hauck et al., 2016). Thus, we expect mineral based OAE to have some impact on marine communities with these impacts being highly dependent on the source minerals used. However, it is important that the environmental impacts and/or co-benefits resulting from OAE are evaluated against the potential climatic benefits.

This study aims to specifically assess and compare the potential implications of silicate- and calcium-based mineral OAE on a coastal plankton community. In order to capture the potential maximum and minimum
acceptable levels of alkalinity enhancement, we increased concentrations of alkalinity in steps of 150 µmol kg$^{-1}$ by 0 to ~600 µmol kg$^{-1}$. Such an increase in alkalinity is expected to influence the phytoplankton community as concentrations of $CO_2$ decrease below previously observed thresholds limiting growth (Chen and Durbin, 1994; Hinga, 2002; Paul and Bach, 2020; Riebesell et al., 1993).

Previous work has identified that increases in alkalinity of 500 µmol kg$^{-1}$ resulted in a significant decrease in
silicate uptake and biogenic silica production (Ferderer et al., 2022). Furthermore, it is well known that silicate has a fertilising effect on diatoms, with concentrations above 2 µmol kg$^{-1}$ often resulting in their dominance within plankton communities (Egge and Aksnes, 1992; Escaravage and Prins, 2002). The dissolution of silicate based minerals such as olivine to enhance alkalinity is predicted to significantly increase silicate concentrations at sites of addition and projected to induce diatom blooms (Hauck et al., 2016). The influence of varying silica
concentrations on diatoms is well known, however the interaction between OAE and enhanced silicate concentrations as a result of silicate based mineral dissolution is yet to be fully explored. Thus, in this study we focus on assessing the influence of mineral based OAE, along an increasing gradient, on the incorporation of silica into the frustules of diatoms (silicification). Our primary goal is to elucidate the potential risks and or co-benefits of mineral based alkalinity enhancement on diatoms.





## 2 Methods

### 2.1 Mesocosm deployment and maintenance

On the 7th of May 2023, ten Kiel Off-Shore Mesocosms for Ocean Simulations (KOSMOS, M1-M10; Riebesell et al., 2013) were deployed from RV ALKOR in Raunefjorden, Bergen, Norway ~1.5 km from the Espegrend marine research field station (Fig. 1). Mesocosms consisted of a cylindrical polyurethane bag 20 m in length, (2 m in diameter, ~60.01 ± 0.01 m³ volume). Mesocosm bags were fitted within 8 m tall floating frames during deployment which were slowly lowered into the fjord, allowing the bags to gently fill while minimising disturbance to the plankton community. Once deployed the mesocosm bags remained open at their base (~20 m) and top (~1 m below the sea surface) allowing water exchange between the mesocosms and fjord. Mesocosms were closed off to the fjord on the 13th of May when divers attached a 2 m long funnel shaped sediment trap to each mesocosm, and the top of each mesocosm bag was raised ~1 m above the surface (Fig. 1). A ring with the same diameter as the mesocosms fitted with a 1 mm mesh was passed through each mesocosm after closing to remove any large nekton or plankton from the mesocosms. The sealing off of the mesocosms from the fjord marked the beginning of the experiment (Day 0). The volume of each mesocosm was determined on day 2 of the experiment via the addition of a NaCl brine solution. Water inside mesocosms was first homogenized by bubbling compressed air up through the mesocosms. Following homogenisation, 50 L of NaCl brine was evenly added to the mesocosm via a bespoke distribution device called "the spider" (Riebesell et al., 2013). The precise addition of NaCl enabled us to calculate the volume of each mesocosm following (Czerny et al., 2013). Mesocosm bags were cleaned approximately every week from the inside and outside to minimise any potential biofouling which may impact the results of the experiment. External cleaning of the mesocosms was conducted by divers and/or surface attendants in small boats using brushes. Internal cleaning of the mesocosms was conducted using a large ring with rubber blades that was the same diameter as the mesocosms. This ring was sunk inside mesocosms with a 30 kg weight attached to its base, removing any growth from the inner walls of the mesocosm bags down to 1 m above the sediment trap.

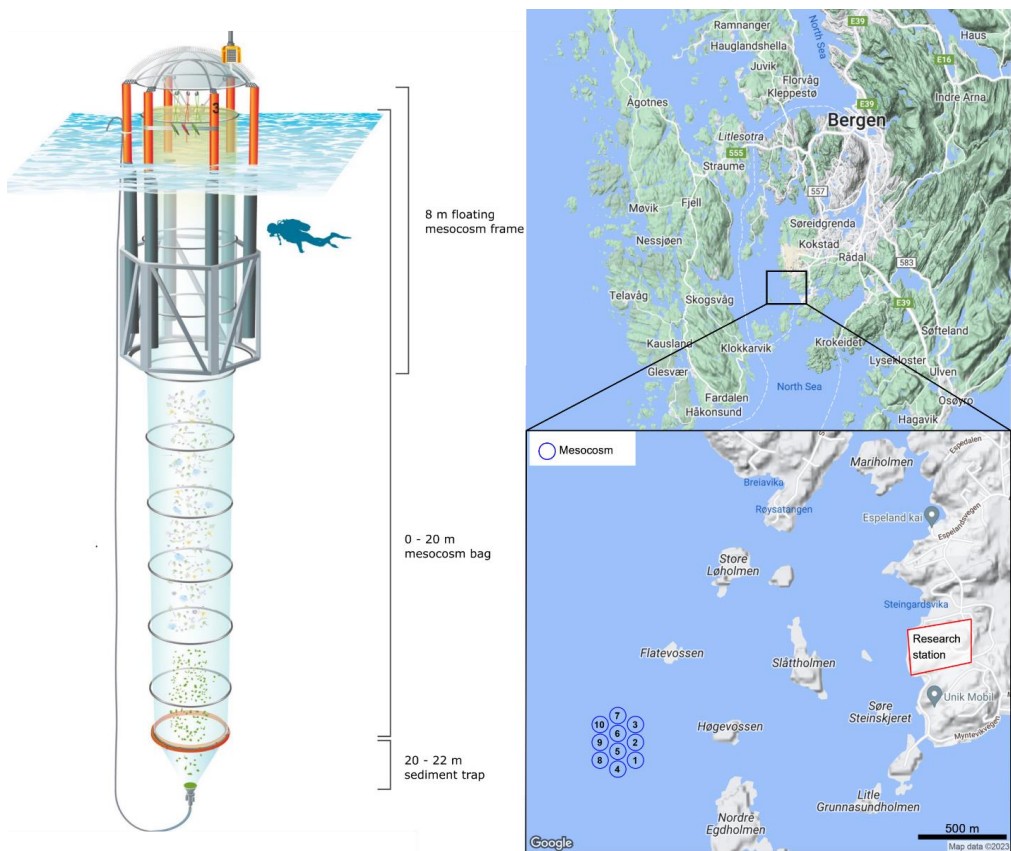

**Figure 1.** Infographic depicting the relevant information pertaining to the mesocosm design and mooring site approximately 1.5 km from the Espegrend Marine research field station in Bergen, Norway (maps produced in Rstudio using Google maps data).

### 2.2 Setup of OAE treatments and nutrient fertilisation

Mesocosms were split into two OAE treatment groups; a calcium-based (Ca-OAE) treatment ($N$ =5) and silicate-based (Si-OAE) treatment ($N$ =5). Alkalinity was enhanced along a gradient in each mineral-based treatment ranging from 0 – 600 µmol kg$^{-1}$ using varying amounts of NaOH (Merck), dissolved in 20 L of Milli-Q®. Simulated differences in the type of OAE were established via the addition of CaCl$_2$·2H$_2$O in the Ca-OAE treatments and MgCl$_2$·6H$_2$O and Na$_2$SiO$_3$·5H$_2$O in the Si-OAE treatments, all dissolved in 20 L of Milli-Q®. The

simulated enhancements of Mg$^{2+}$ and Ca$^{2+}$ were proportional to the addition of NaOH, i.e., increases by half the alkalinity enhancement. In contrast Na$_2$SiO$_3$ was increased by equal concentrations (target of 75 µmol L$^{-1}$) in all mesocosms within the Si-OAE treatment group (including the control), instead of a gradient from 0 - 150 µmol L$^{-1}$, which would be the corresponding concentrations for olivine dissolution. This approach was adopted due to metasilicate solubility restrictions (data not shown), the potential for colloid formation to occur at high

concentrations and enable clear distinctions to be made between silicate and TA effects. Finally, the increase in



TA from silicate additions in a 2:1 ratio was taken into account by reducing respective NaOH additions and the addition of HCl in the silicate-based control ($\Delta$TA = 0 µmol kg$^{-1}$).

At the time of closure, all mesocosms had low concentrations of macronutrients (0.10 ± 0.019 µmol L$^{-1}$ NO$^-_3$, 0.03 ± 0.005 µmol L$^{-1}$ PO$^{3-}_4$ and 0.16 ± 0.048 µmol L$^{-1}$ Si(OH)$_4$). After observing communities in a prolonged phase of oligotrophic conditions, macronutrients were added to the mesocosms on day 26 (final range across mesocosms = 3.59 – 3.8 µmol L$^{-1}$ NO$^-_3$, 0.19 – 0.24 µmol L$^{-1}$ PO$^{3-}_4$ and 0.39 – 1.03 µmol L$^{-1}$ Si(OH)$_4$) to stimulate the phytoplankton community. Macronutrients were added to the mesocosms as two separate 20 L solutions with one consisting of NaNO$_3$ (Merck, > 99.5%) and Na$_2$HPO$_4$. H$_2$O (Merck, > 99.5%) and the other consisting of Na$_2$SiO$_3$.5H$_2$O (Roth > 95%). Inorganic nutrient concentrations were measured the day before nutrient additions and ~2 hours after the addition of nutrients to quantify the additions and ensure appropriate stoichiometry within mesocosms. After the addition of nutrients, it was noted that the stoichiometry of macronutrients was not even across mesocosms, later identified to have been the result of a mistake during solution preparation. As such, a second addition of nitrate was completed on day 28 for those mesocosms below target concentrations. All solutions were added homogeneously to mesocosms using the spider distribution device.

Given the additions of alkalinity on day 7 and macronutrients on day 26 and 28, the experiment was divided into three distinct phases; phase 0 representing the period prior to alkalinity enhancement (day 0 – 6), phase I representing conditions prior to the addition of macronutrients (day 7 - 28) and phase II representing the period after nutrient additions (day 29 – 54).

### 2.3 Sampling methods

Sampling of the mesocosms was conducted every second day from small boats with sediment sampling first (0800 – 1000; here and in the following GMT +2) followed by particulate and dissolved substance sampling (0900 – 1300), zooplankton sampling (1000 – 1300) and finally CTD, FastOcean APD/fluoroprobe (1400 – 1600). With the exception of particulate and dissolved substance sampling, which was carried out at random, mesocosms were sampled in order from M1 through to M10 with fjord samples taken directly next to M5. Sample containers were stored in boxes to avoid excess light and heat exposure during sampling and upon return to the research station (directly after each round of sampling) were transferred to a room at ambient water temperature (8.7 – 15.4 ºC) until further processing. The sampling schedule remained consistent with the exception of day 15 where only sediment sampling was undertaken due to unsafe weather conditions. Additional samples for the determination of dissolved inorganic nutrients were taken on day 26 and day 28 to assess stoichiometry post nutrient additions performed earlier the same day.

Sinking particles were collected from the sediment traps of each mesocosm via a silicon tube attached to the base of the sediment trap at one end and a manual vacuum pump at the surface (Boxhammer et al., 2016). Suspended particulate matter and dissolved substances were collected using 5 L integrated water samplers (IWSs; Hydro-Bios, Kiel). IWSs were equipped with pressure sensors enabling an even collection of water within a specified depth, from the surface to the top of the sediment trap (0 – 20 m). Four IWSs were taken within each mesocosm and fjord which were transferred into 10 L polyethylene carboys. Samples for quantification of changes in



carbonate chemistry were collected from the first IWS taken within each mesocosm and filled directly into 500
ml glass bottles following protocols outlined in SOP 1 from (Dickson et al., 2007).

### 2.4 Carbonate chemistry and dissolved inorganic nutrients

Samples for total alkalinity (TA), pH and dissolved inorganic nutrients were sterile filtered using a peristaltic

pump and 25 mm, 0.2 µm pore size, PES membrane, syringe filters to minimise biological processes and remove

particles which can influence respective analyses. Dissolved inorganic nutrient concentrations $NO_3^-$, $NO_2^-$, $PO_4^{3-}$,

and $Si(OH)_4$ were determined spectrophotometrically following methods outlined by (Hansen and Koroleff,

1999). Dissolved inorganic nutrient samples were measured in triplicate to control for technical variability

between measurements across the experiment. TA was determined using a two-step open cell titration following

SOP3b outlined by (Dickson et al., 2007). TA samples were measured in duplicate on a Metrohm 826 Compact

Titrosampler coupled with an Aquatrode Plus with PT1000 temperature sensor and calibrated against certified

reference material (CRM batch 93) supplied by Prof. Andrew Dickson's laboratory. pH was determined in

duplicate via spectrophotometric methods outlined in (Dickson et al., 2007) (not shown here).

### 2.5 Particulate matter analysis

Sediment trap samples were processed immediately upon return of the sampling boat to the research station.

Sample weight was first determined gravimetrically before resuspension of particles and homogenisation of the

sample for subsampling (e.g., dissolution assays, particle sinking velocity). The remaining sample was enriched

with 3M $FeCl_3$ followed by 3M NaOH to enhance flocculation, coagulation and subsequent sedimentation of

particles while maintaining pH (Boxhammer et al., 2016). Approximately 1 hour after settling the supernatant was

removed and samples centrifuged in two steps: first for 10 min at 5200 g in a 6-16KS centrifuge (Sigma) and then

again for 10 min at 5000 g in a 3K12 centrifuge (Sigma). Following each step any supernatant was removed and

the remaining pellet freeze dried to remove residual moisture. Finally, the dried samples were pulverised into a

homogenous powder using a cell mill and were transported to GEOMAR, Kiel, Germany for further analysis.

Subsamples of the powder used to determine concentrations of biogenic silica (BSi) were placed in 60 ml

Nalgene™ polypropylene bottles, filled with 25 ml of 0.1 M NaOH solution, and then placed in a shaking water

bath at 85 ºC. After 135 min the bottles were removed and cooled before the addition (25 ml) of 0.05 M $H_2SO_4$ to

stop the leaching processes. The concentration of dissolved silicate was then measured spectrophotometrically

following (Hansen and Koroleff, 1999). BSi concentrations of the measured subsamples were then scaled to

represent the total sample from sediment traps, normalised to mesocosm volume and are reported as the

accumulation of BSi in the sediments over the experimental period.

Analysis of major elemental pools and phytoplankton pigments in the water column subsamples (pre-filtered

through a 200 µm screen) of 0.5 – 1 L were taken from carboys after gentle mixing to homogenise samples.

Subsamples for BSi were filtered onto cellulose acetate filters (pore size 0.45 µm) using a vacuum filtration system

at ≤ 200 mbar and stored in plastic vials at -20 ºC until analysis the following day. Filters were then placed in 60

ml Nalgene™ bottles, digested, and analysed following the same methods for BSi in the sediments described

above. Chlorophyll *a* was filtered onto glass fibre filters (GFF, nominal pore size = 0.7 µm) while minimising





light exposure. Immediately after filtration filters were stored in plastic vials at -80 ⁰C until analysis the following

day. Samples were extracted with 90% acetone and homogenised using glass beads in a cell mill. After

homogenisation samples were centrifuged (10 min 800 g, 4ºC), then the supernatant was removed and analysed

on a fluorometer (Turner 10-AU) to determine Chl *a* concentrations (Welschmeyer, 1994).

### 2.6 PDMPO labelling and determination of silicification rates via fluorescent microscopy

To investigate differences in diatom silicification, 350 ml samples were collected by means of IWS from each

mesocosm every two to four days (variation in sampling schedule occurred due to unforeseen circumstances such

as extreme weather events and COVID-19 infections). Due to the low abundance of diatoms within mesocosms,

(as observed through microscopy and BSi concentrations) samples were gravimetrically concentrated from 350

ml to 70 ml using a 47 mm filtering apparatus and polycarbonate filters (3 µm pore size). Organisms were

resuspended by gentle stirring and inversion of the filtration system before the sample was transferred to three 17

ml            polycarbonate            tubes.            The            fluorescent            dye            2-(4-pyridyl)-5-((4-(2-

dimethylaminoethylaminocarbamoyl)methoxy)phenyl)oxazole (PDMPO; LysoSensor yellow/blue DND-160

from ThermoFisher Scientific) was added at a final concentration of 0.125 µM before samples were incubated in

a flow-through incubator placed outside of the Espegrend Marine research field station for temperature control

(inlet location was ~1.5 km from the mesocosms so that temperature in the incubator was similar to mesocosms).

The incubator was screened with shade cloth to ~30% incident irradiance and incubations lasted for ~24 hours.

At the conclusion of the incubation, 15 ml of the sample was filtered under gentle vacuum pressure onto 25 mm,

0.8 µm black polycarbonate membrane filters. Filters were mounted onto microscope slides with a drop of Prolong

gold antifade followed by a glass coverslip and sealed with clear nail varnish. Prepared slides were then stored in

the dark at -20 ºC for later analysis via fluorescent microscopy at the University of Tasmania, Australia (within 6

months).

Prepared microscope slides were imaged using the software NIS-elements and a Nikon eclipse Ci microscope

equipped with a UV-1A (longpass) filter cube, Nikon DS-Ri2 camera and Mercury lamp (Nikon C-SHG1). Prior

to analysing slides, a yellow fluorescence slide (Thorlabs FSK3) was imaged ten times and the average

fluorescence subtracted from all images taken that day to account for variation in the mercury lamp across imaging

days. The entirety of each filter was systematically scanned at ×200 magnification, when a cell was located, it was

imaged at ×400 magnification. Cells were identified to genus level as brightfield imaging was not possible on the

black polycarbonate filters and fluorescent images did not provide enough detail for accurate identification beyond

this level. Images were later analysed by quantifying single cell fluorescence following a custom-made procedure

in ImageJ on the original TIFF images. For each image, cell/s were selected so that minimal background area was

included before the fluorescence of the selected cell was recorded. Background fluorescence was measured at four

locations directly surrounding the cell where no other cells were present, and the average background fluorescence

subtracted from cell fluorescence. Total cell fluorescence, corrected for background fluorescence, was then

normalised to cell area to give mean cell fluorescence. Due to low abundances of diatoms and thus insufficient

counts for meaningful analyses, only days 7, 11 and 17 were analysed prior to the addition of nutrients on day 26



and 28. After the addition of nutrients all filters were analysed, however due to an outbreak of COVID-19, samples for the determination of silicification could not be taken for the final eight days of the experiment.

Due to technical issues and the unavailability of specific instrumentation, we were unable to measure total community fluorescence during the experiment and as such convert fluorescence values to BSi incorporation. However, the aim of this experiment was to identify any relative differences in taxa specific rates of silicification, something which is not achievable through the measurement of BSi. This was achieved with the use of the fluorescent dye PDMPO which is incorporated at a rate proportional to BSi incorporation in diatoms (Leblanc and Hutchins, 2005; Mcnair et al., 2015; Znachor and Nedoma, 2008). PDMPO uptake and subsequent fluorescence

within diatom cells therefore provides an appropriate proxy for the incorporation of silica into newly formed frustules irrespective of the units. As such the PDMPO-fluorescence of cells measured here as the fluorescence of a given cell normalised to cell area, is referred to as silicification throughout this text.

### 2.7 Statistical analysis

To explore the effect of the alkalinity source mineral and alkalinity enhancement across mesocosms we first

visualised the dataset using non-metric multidimensional scaling (nMDS) plots. Three separate plots were produced to explore the effects of (a) the treatments over the total extent of the experiment, (b) prior to the addition of nutrients, and (c) post nutrient addition on silicification.

Linear mixed-effects models were used to quantify the influence of the treatments (total alkalinity and alkalinity

source mineral) on silicification. A model was first run at the community level with alkalinity source mineral, total alkalinity, phase, and diatom genus as fixed effects, and silicification (square root transformed) as the dependent variable. Due to the significant variation in silicification attributed to genus, individual models were carried out for each diatom genus with alkalinity source mineral, phase, and total alkalinity as fixed effects and silicification (square root or log transformed) as the dependent variable. To account for temporal pseudo-

replication in the models, mesocosm ($N = 10$) was nested within sampling occasion (day) and fitted as a random effect in all models. Several linear mixed-effects models were fit for the community and each genus, with non-significant interactions removed and Akaike Information Criteria (AIC) used to determine the best model in each circumstance. In order to control for temporal autocorrelation in the models an autocorrelation structure of order one at the level of random effect was also included.


Finally linear models were used to assess the influence of total alkalinity and alkalinity source mineral on the concentration of BSi in the water column and accumulation of BSi in the sediments. Linear models were run for each phase of the experiment with mean water column BSi and mean accumulated sediment BSi over each phase fitted as dependent variables and mean total alkalinity and alkalinity source mineral as fixed effects. An additional

linear model was run to assess total accumulated BSi in the sediment trap with the accumulated sediment BSi up until day 53 fitted as the dependent variable and total alkalinity and alkalinity source mineral fitted as fixed effects. All statistical analyses including nMDS plots were conducted in Rstudio v 2023.6.1.524 (Posit Team (2023), 2023).



## 3 Results

Chlorophyll $a$ concentrations were relatively low at the beginning of the experiment with $1.01 \pm 0.17\,\mu g\,L^{-1}$ (mean $\pm$ SD) on day 3 (Fig 2d). Concentrations of $NO_3^-$ were below detection limit, thereby constraining phytoplankton growth during phase I of the experiment (mean $NO_3^-$ day 7 – 25 = $0.004 \pm 0.035\,\mu mol\,L^{-1}$) (Fig 2a). In contrast there was residual $PO_4^{-3}$ ($0.021 \pm 0.022\,\mu mol^{-1}$) and $Si(OH)_4$ (Ca-OAE treatment = $0.202 \pm 0.99\,\mu mol^{-1}$, Si-OAE treatment = $67.929 \pm 1.04\,\mu mol^{-1}$) which likely supported the phytoplankton community in utilising remineralised

nitrogen until the addition of nutrients on day 26 and 28 (Fig 2b, 2c). Although ~70 $\mu mol\,L^{-1}$ of $Na_2SiO_3$ was added to the Si-OAE treatment, there was no discernible depletion of $Si(OH)_4$ during phase I (Fig 2c). The addition of macronutrients ($NO_3^-$, $PO_4^{-3}$, and $Si(OH)_4$) can be seen on day 26, with a secondary addition on day 28 to correct for unwanted differences in the stoichiometry between mesocosms (Fig 2). During phase II, nutrients steadily declined until the majority was depleted between days 39 and 49 (Fig 2). Chlorophyll $a$ increased rapidly

after day 35 with the exception of M10 (Si-150) that exhibited an exponential increase after day 33 (Fig 2d). The slow increase in chlorophyll $a$ was likely due to the prolonged nutrient deficit within the mesocosms and subsequent small seed population. Mesocosms in the Si-OAE treatment appeared to exhibit slightly higher and somewhat earlier peaks in Chlorophyll $a$ when compared to the calcium-based treatment (Fig. 2c, phase II). This trend is supported by the uptake of $NO_3^-$ with mesocosms in the Si-OAE treatment, exhibiting marginally faster

depletion of $NO_3^-$ in comparison to the Ca-OAE treatment. There was no discernible relationship between total alkalinity and chlorophyll $a$, $NO_3^-$ or $PO_4^{-3}$ observed across the extent of the experimental period or in a particular phase (Fig 2). However, in the Si-OAE treatments concentrations of $Si(OH)_4$ were noticeably lower in mesocosms with high alkalinity (Fig 2c). This trend appeared directly after the addition of the treatments early in the experiment and was conserved until the end of the study (Fig 2c).






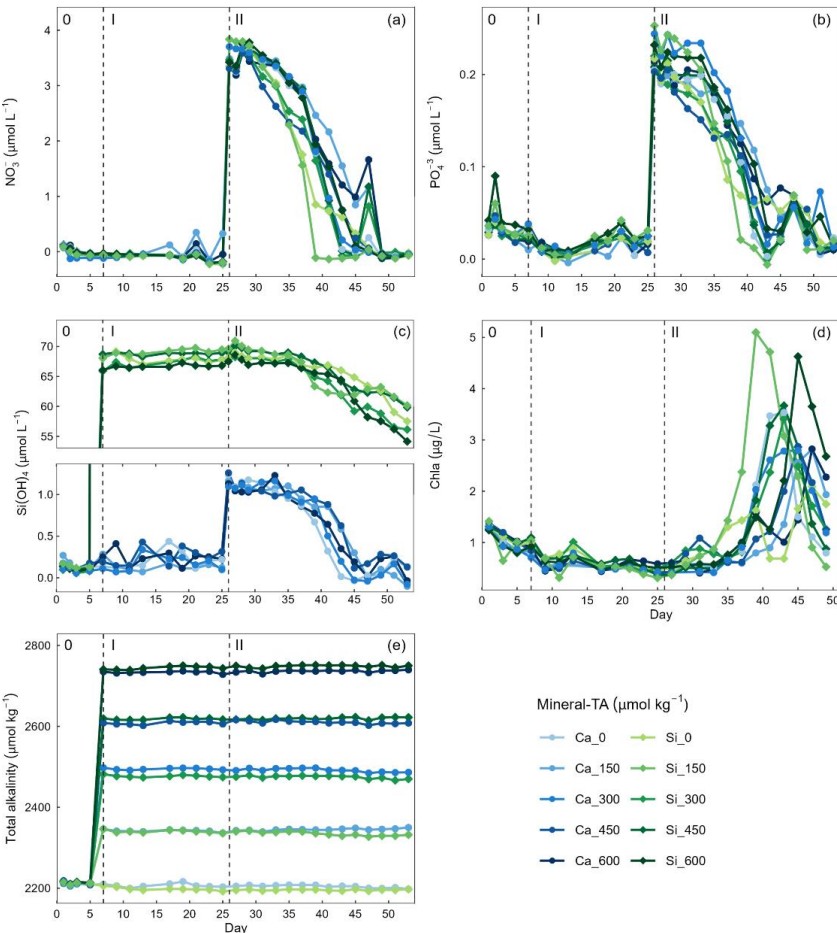

**Figure 2.** Temporal variation in dissolved inorganic nutrients **(a)** nitrate ($NO_3^-$), **(b)** phosphate ($PO_4^{-3}$), **(c)** silicate ($Si(OH)_4$), **(d)** chlorophyll $a$ (Chla) and, **(e)** total alkalinity. Dissolved inorganic nutrient measurements commenced on day 0 while chlorophyll $a$ measurements commenced on day 3. Vertical dashed lines represent the respective phases of the experiment, phase 0 (pre-alkalinity enhancement), phase I (pre-nutrient addition), phase II (post-nutrient addition).

Nonmetric multidimensional scaling (nDMS) (Fig. 3) revealed distinct distances among treatments, including different alkalinity source minerals and total alkalinity, in relation to silicification of the various diatom genera, although relatively small distances are observed between *Pseudo-nitzschia* and *Nitzschia*, likely due to their morphological similarities and therefore similar silica content. The distances between polygons, representing the Si- and Ca-based mineral treatments, indicate differences in silicification between the Si- and Ca-based mineral treatments over the total experimental period and after the addition of nutrients (Fig. 3a,c). In contrast, during phase I (pre-nutrient addition) polygons representing the Si- and Ca-based mineral treatments are overlapping, suggesting a weak relationship between alkalinity source mineral (Ca or Si) and silicification prior to the addition of nutrients (Fig. 3b). In all plots, symbols representing the differing levels of total alkalinity within the Si-based

treatment are relatively closer, when compared to the Ca-based treatment which exhibits greater spread between differing levels of total alkalinity.

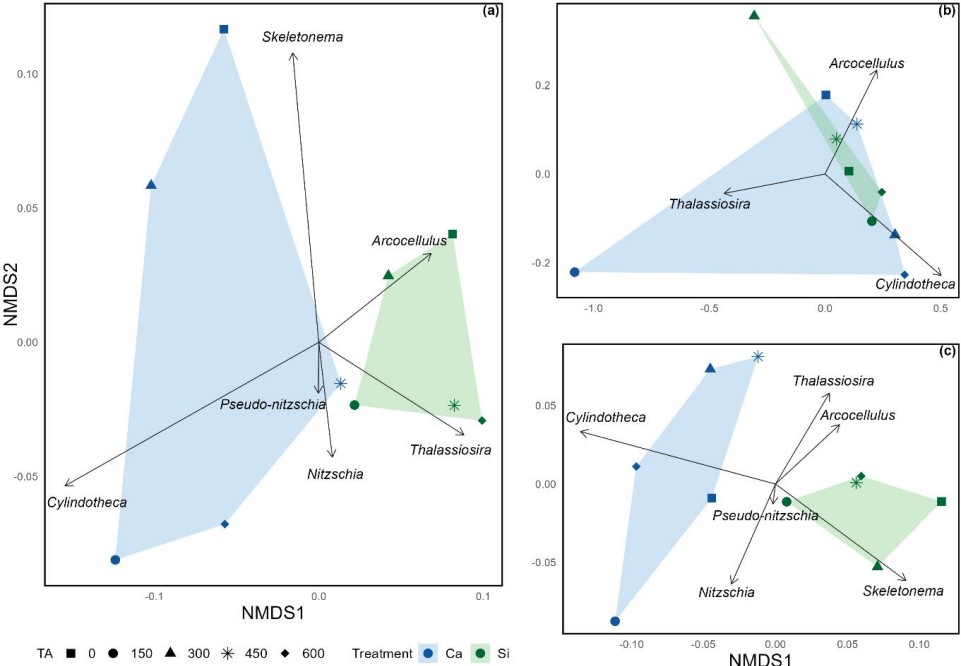


**Figure 3.** Nonmetric multidimensional scaling ordination exploring the mean silicification of diatom genera across **(a)** the complete extent of the mesocosm experiment (stress = 0.0502), **(b)** pre-nutrient addition, phase I (stress = $3.62e^{-5}$), and **(c)** post nutrient addition, phase II (stress = 0.0612). Due to the low stress values obtained (<0.20), it is assumed that all configurations accurately represent distinct dissimilarities in silicification among 330 diatom genera.

### 3.1 Results of linear mixed effects models

Analysis of community silicification supported the distances observed within the nMDS plots with alkalinity source mineral (Ca or Si) having a significant influence on the silicification of diatom cells (Table 1, Fig. 4). Cells exposed to the Si-OAE treatment were more heavily silicified irrespective of changes in total alkalinity (Fig. 4). 335 However, the significant interaction between alkalinity source mineral and genus, indicates that the effect of the alkalinity source mineral on silicification varies between genera (Table 1). In contrast, total alkalinity did not exhibit a significant influence on diatom silicification, despite visual trends of increasing silicification with total alkalinity observed on select days (day 7 and 11, Fig. 4). Nevertheless, the significant interaction between genus and total alkalinity suggests the influence of total alkalinity on silicification varies between genera. Additionally, 340 there was a significant difference in community silicification between the two experimental phases, suggesting that changes in macronutrient concentrations (nitrate and phosphate) impacted silicification (Table 1 and Figure





4). Due to the significant interactions between genus and the treatments (alkalinity source mineral and total alkalinity) further investigation was conducted via genus level models.

**Table 1.** Statistical results of linear mixed-effects model assessing the influence of mineral based OAE on diatom community silicification

| Source of variation | *df* | *F-value* | *P-value* |
|---|---|---|---|
| **Alkalinity source mineral (Ca or Si)** | 1,68 | 22.67 | <0.001* |
| **Genus** | 5,11335 | 887.23 | <0.001* |
| **Total alkalinity** | 1,68 | 0.024 | 0.88 |
| **Phase** | 1,6 | 10.10 | 0.02* |
| **Alkalinity source mineral*Genus** | 5,11335 | 24.12 | <0.001* |
| **Total alkalinity*Genus** | 5,11335 | 17.39 | <0.001* |

*Genus was included within the model as it explained a significant proportion of variance allowing for correct model fitting.  *P<0.05*

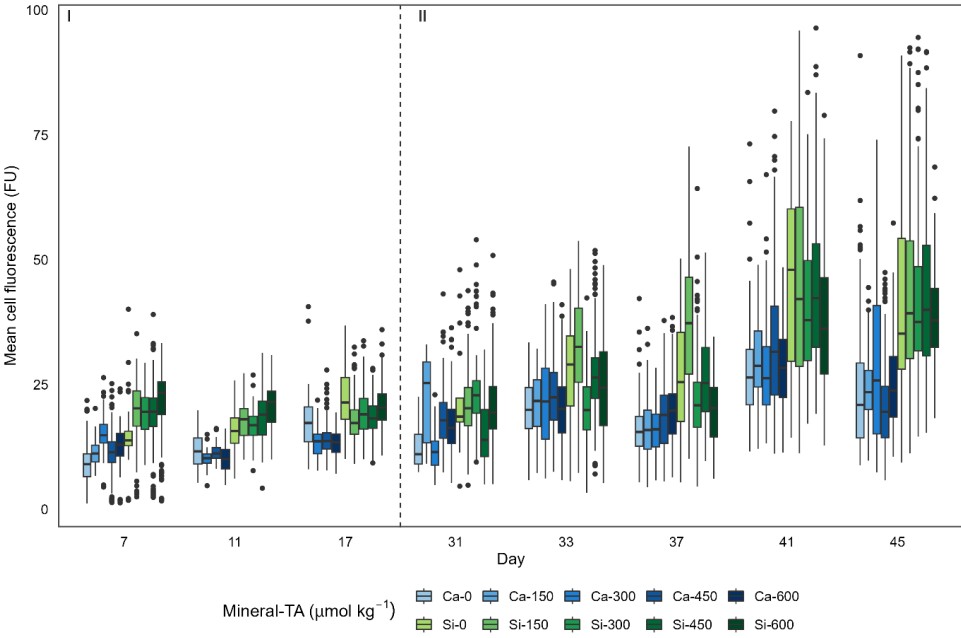

**Figure 4**. Single cell silicification of the diatom community depicted as mean fluorescence (PDMPO) normalised to cell surface area and reported in fluorescence units (FU). Data visualised as box plots, with colours representing the different mineral sources (Ca-OAE; blue and Si-OAE; green) and shading indicating the total alkalinity gradient with darker colours indicating higher total alkalinity. Nutrient addition on day 26 and 28 is represented by the dashed line dividing the experiment into phase I (pre-nutrient addition) and phase II (post-nutrient addition).





Genus specific models revealed no significant differences between the two nutrient phases and/or alkalinity treatments with the exception of *Pseudo-nitzschia,* which displayed a significant increase in silicification with increasing alkalinity. Each genus was significantly influenced by the alkalinity source mineral (Ca or Si) with

those exposed to the Si-OAE treatment exhibiting greater silicification (Table 2, Fig. 5). This is with the exception of *Cylindrotheca* which showed no significant relationship between silicification and alkalinity source mineral, total alkalinity and/or phase (Table 2, Fig. 5).

**Table 2.** Statistical results of linear mixed effects models exploring the effect of alkalinity source mineral, total

alkalinity, and experimental phase on silicification for a given genus.

| Genus | Source of variation | df | F-value | P-value |
|---|---|---|---|---|
| *Arcocellulus* | **Alkalinity source mineral** | 1,59 | 14.09 | <0.001* |
| | **Total alkalinity** | 1,59 | 0.64 | 0.43 |
| | **Phase** | 1,6 | 3.13 | 0.13 |
| *Cylindrotheca* | **Alkalinity source mineral** | 1,38 | 0.15 | 0.70 |
| | **Total alkalinity** | 1,38 | 4.06 | 0.05 |
| | **Phase** | 1,6 | 5.26 | 0.06 |
| *Nitzschia* | **Alkalinity source mineral** | 1,36 | 8.61 | 0.006* |
| | **Total alkalinity** | 1,36 | 0.47 | 0.50 |
| *Pseudo-nitzschia* | **Alkalinity source mineral** | 1,56 | 50.5327 | <0.001* |
| | **Total alkalinity** | 1,56 | 5.5192 | 0.02* |
| | **Phase** | 1,6 | 5.2467 | 0.06 |
| *Skeletonema* | **Alkalinity source mineral** | 1,53 | 23.21 | <0.001* |
| | **Total alkalinity** | 1,53 | 2.11 | 0.15 |
| | **Phase** | 1,6 | 4.80 | 0.07 |
| *Thalassiosira* | **Alkalinity source mineral** | 1,53 | 68.64 | <0.001* |
| | **Total alkalinity** | 1,53 | 3.84 | 0.06 |
| | **Phase** | 1,6 | 4.08 | 0.09 |

*Phase was not included for Nitzschia as this genus was only present in phase II. * P<0.05*





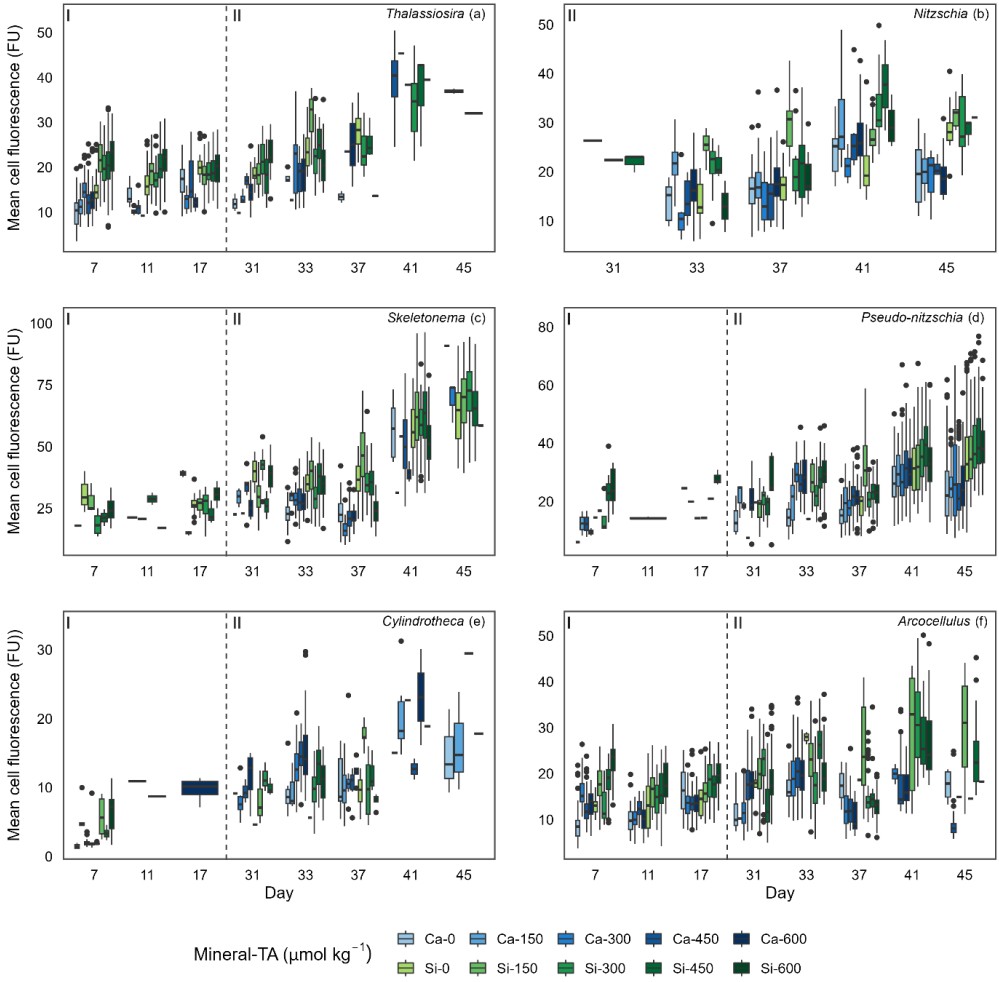

**Figure 5.** Boxplots depicting single cell silicification of different genera. Silicification is shown as mean cell fluorescence (PDMPO) normalised to cell area and reported in fluorescence units (FU). Colours representing the different mineral sources (Ca-OAE; blue and Si-OAE; green) and shading indicating the total alkalinity gradient with darker colours indicating higher total alkalinity. Nutrient addition on day 26 and 28 is represented by the dashed line dividing the experiment into phase I (pre-nutrient addition) and phase II (post-nutrient addition), with genera names in the top right of each plot.

Concentrations of BSi in the water column can be seen decreasing from day 0 (mesocosm closure) and remaining low until day ~33 (Fig 6a). Directly after the addition of $Na_2SiO_3$ (day 7), BSi in the water column spiked for one day in the silicate-based treatments, while no increase was observed in the calcium-based treatment (Fig 6a). BSi began to increase in all mesocosms between days 33-35, however concentrations in the Ca-OAE treatments remained relatively low (<2 $\mu mol\ L^{-1}$) for the extent of the experiment (Fig 6a). We observed no significant relationship between BSi and alkalinity in either of the alkalinity source mineral treatments (Ca or Si) or any phase





of the experiment (Table 3). During phase II, concentrations of BSi in the water column were significantly higher in the Si-OAE treatment when compared to the Ca-OAE treatment (Fig 6a). Additionally, we observed significant differences in the accumulation of BSi in the sediments between the two alkalinity source mineral types (Table 3, Fig. 6b). Similar to the water column, there was an initial increase in sedimented BSi in the Si-OAE treatment when compared to the Ca-OAE treatment (Fig 6b). Visual inspection of Figure 6b suggests that BSi in the sediment trap increased with alkalinity, however this was not statistically significant for any phase of the experiment (Table 3).

**Table 3.** Results of linear models exploring the effect of alkalinity source mineral and total alkalinity on average concentrations of BSi in the water column or sediment trap for a given phase.

| Phase | Parameter | Source of variation | df | F-value | P-value |
|---|---|---|---|---|---|
| **Phase 0** | **BSi water column** | **Mineral** | 1 | 0.70 | 0.43 |
| | | **TA** | 1 | 0.70 | 0.43 |
| **Phase 0** | **BSi sediment** | **Mineral** | 1 | 0.20 | 0.67 |
| | | **Total alkalinity** | 1 | 0.35 | 0.57 |
| **Phase I** | **BSi water column** | **Mineral** | 1 | 70.88 | <0.001* |
| | | **TA** | 1 | 1.76 | 0.23 |
| **Phase I** | **BSi sediment** | **Mineral** | 1 | 46.44 | <0.001* |
| | | **Total alkalinity** | 1 | 4.80 | 0.06 |
| **Phase II** | **BSi water column** | **Mineral** | 1 | 47.15 | <0.001* |
| | | **TA** | 1 | 2.02 | 0.20 |
| **Phase II** | **BSi sediment** | **Mineral** | 1 | 82.6109 | <0.001* |
| | | **Total alkalinity** | 1 | 0.64 | 0.45 |
| **Day 53** | **BSi sediment** | **Mineral** | 1 | 45.57 | <0.001* |
| | | **Total alkalinity** | 1 | 0.0005 | 0.98 |

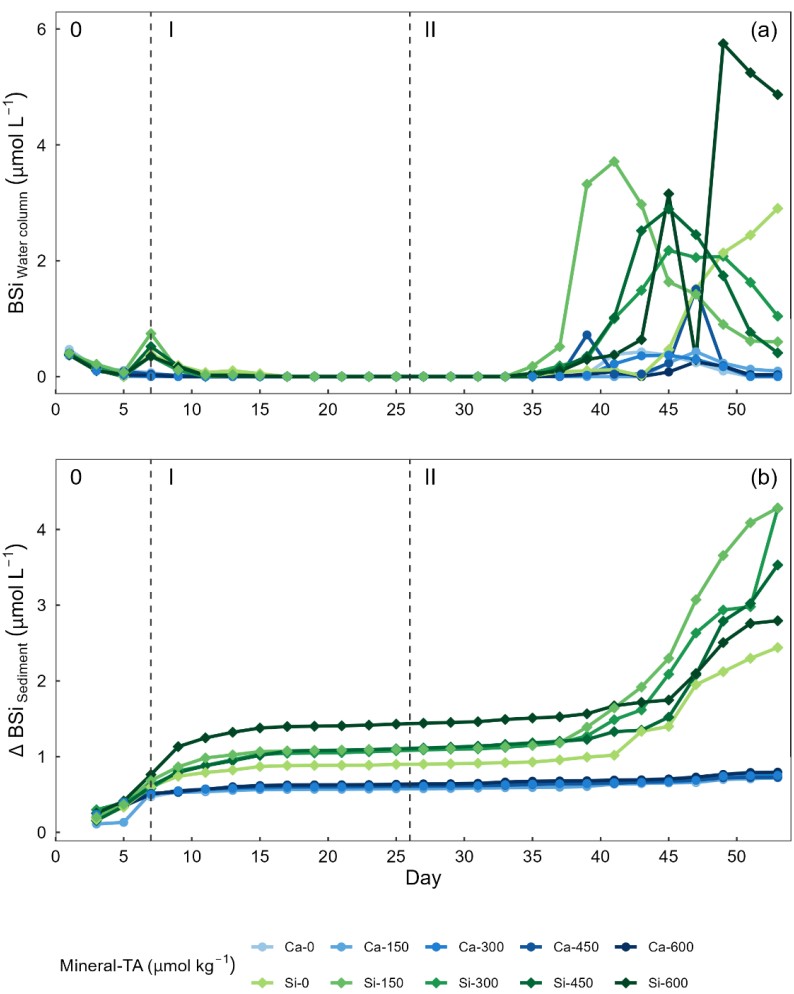

**Figure 6.** Temporal variations of **(a)** biogenic silica (BSi) in the water column, and **(b)** accumulation of BSi in
the sediment trap across alkalinity source minerals and total alkalinity treatments during the extent of the
experimental period. Nutrient addition on day 26 and 28 is represented by the dashed line dividing the experiment
into phase I (pre-nutrient addition) and phase II (post-nutrient addition).

## 4 Discussion

Understanding the potential environmental implications of CDR methods such as OAE is a crucial step before
decisions are made upon their implementation at large scales. The aim of this mesocosm study was to form part
of this research by assessing the potential effects of calcium- and silicate-based mineral OAE on a coastal plankton
community. Here, we specifically discuss the influence of simulated mineral based OAE on diatom community
and genus specific silicification. The results of our study revealed silicate based OAE to significantly increase
silicification in the diatom community and all genera with the exception of *Cylindrotheca*. This trend was



confirmed by increased concentrations of BSi in the water column and accumulated in the sediment trap of mesocosms in the Si-OAE treatment. The increase of seawater alkalinity by 0 – 600 µmol kg⁻¹ had no effect on the concentration of BSi in the water column, accumulated in the sediment or diatom community silicification. *Pseudo-nitzschia* was the only genus significantly affected by increases in alkalinity, exhibiting a significant increase in silicification with increasing alkalinity. In conjunction with published OAE research, our findings

highlight the need for research to cover a broad range of environmental conditions, approaches to OAE and marine communities.

### 4.1 Temporal dynamics of biogenic silica in the water column and sediment

We observed no significant relationship between alkalinity and BSi in the water column or accumulated in the sediments (Fig 6 a,b). In contrast, concentrations of BSi in the water column and accumulated in the sediments

were significantly greater in the Si-OAE treatment before and after macronutrient additions. These trends support observations for community- and genera-specific silicification, with diatoms in the Si-OAE being more heavily silicified.

Notably, BSi accumulation in the sediments of the Si-based OAE treatment group was greatest in the highest

alkalinity mesocosm (Δ600 µmol kg⁻¹) and lowest in the control mesocosm (Δ0 µmol kg⁻¹) during phase I (Fig. 6b). This difference emerged immediately after the addition of $Na_2SiO_3$ to the Si-based OAE treatment. However, no build-up of chlorophyll $a$ or significant build-up of BSi in the water column was observed in the days prior to the emergence of this trend, suggesting the sedimented BSi is an artefact of the $Na_2SiO_3$ addition to the silicate-based treatments. Our interpretation of these findings are that (1) there were residual precipitates in the $Na_2SiO_3$

solution added to the mesocosms resulting in increased BSi in the water column, and/or (2) that there was pH dependent, inorganic precipitation of amorphous silicate, with these precipitates sinking out into the sediment trap (Goto, 1956; Okamoto et al., 1957; Owen, 1975). The latter is supported by recent work conducted by Gately et al. (2023), whose abiotic experiments revealed a decrease in DSi as TA increased, with scanning electron microscopy revealing mineral precipitates formed in high alkalinity seawater to be primarily composed of silicon

and oxygen. Precipitation of silica within mesocosms may have been supported by ionic interactions between magnesium (or trace metals e.g. iron, aluminium) and silicate, pressure and relatively low temperatures at depth (Ehlert et al., 2016; Goto, 1956; Spinthaki et al., 2018).

### 4.2 Effect of enhanced silicate concentration on silicification rates

Ca- and Si-based OAE appeared to have a notable influence on silicification of the diatom community. However,

this influence is attributed to the mineral treatment type, either silicate or calcium based, with a significant relationship between silicification and alkalinity only observed for the genus *Pseudo-nitzschia*. Diatoms in the Si-OAE treatment incorporated considerably more silica over the 24-hour incubation period resulting in increased silicification. This outcome was not unexpected, especially considering the consistently low (likely limiting) concentrations of $Si(OH)_4$ observed in the Ca-OAE treatment throughout the majority of the experiment. $Si(OH)_4$

is the key nutrient in the construction of the silicate-based frustule of diatoms, with low concentrations often becoming a limiting factor for growth (Martin-Jézéquel et al., 2000). It has been shown that before silicate concentrations become growth limiting, diatoms first respond by thinning their frustules (McNair et al., 2018;





Paasche, 1975). Whilst this phenomenon has been observed in several studies (McNair et al., 2018; Rocha et al., 2010; Shimada et al., 2009), our mechanistic understanding of how diatoms adjust their silicon quotas is unclear

(Milligan et al., 2004). Interestingly, an initial thinning followed by subsequent thickening of diatom frustules has been observed at non-growth limiting silicate concentrations (McNair et al., 2018). This may be an adaptive trade-off by which diatoms respond to lower silicate concentrations by decreasing their silicon quotas in favour of maintaining similar growth rates; a strategy that allows cells to respond to dynamic changes in silica concentrations while maintaining a similar population size. Such rapid responses have been observed in both

culture and field based experiments with diatoms responding to increases in silica within several hours while responses to nitrate additions, after prolonged nitrate stress, took over 30 hrs (McNair et al., 2018; Rocha et al., 2010). As such, it is possible that the low concentrations of $Si(OH)_4$ observed in the Ca-OAE treatment, resulted in diatoms prioritising growth over silica incorporation leading to significantly less silicification when compared to the Si-OAE treatment.


Interestingly, the experimental phase (pre- or post-nutrient addition) did not influence the relationship between silicification and mineral source (Si or Ca). This is supported by additional statistical analysis assessing the period at which nutrients were replete (days 28 – 39) revealing no significant difference in community silicification between the Ca- and Si-OAE treatments (Table A1). Hence, we may infer that the silicate-based treatment only

benefited the diatom community through the increase in silicification during periods where otherwise $Si(OH)_4$ would be limiting. Nevertheless, changes in the normal cycling of silicate, resulting in increased silicification is likely to have significant ecological implications including shifts in community composition and predation rates (Egge and Aksnes, 1992; Liu et al., 2016).

We observed a significant relationship between phase and community silicification suggesting that the low $NO_3^-$ and $PO_4^{-3}$ concentrations across all mesocosms during phase I of the experiment, resulted in decreased silicification, irrespective of $Si(OH)_4$ concentrations. Increases in cell silica quotas have often been correlated with decreased growth rates and subsequent lengthening of the G2 phase allowing more silica to be incorporated before cell division (Claquin et al., 2002; Martin-Jézéquel et al., 2000; Timmermans et al., 2004). Furthermore,

decreased nitrate, phosphate and light availability have all been shown to limit diatom growth and subsequently increase silicification (Claquin et al., 2002; Martin-Jézéquel et al., 2000). As such, if growth rates were nutrient limited during phase I, we would expect an increase in silicification. As this was not the case in our experiment we suggest, whilst nutrients may have been growth-limiting, they were not low enough to induce the observed changes in community level silicification across experimental phases. Instead, our observations are likely

explained by a nutrient-induced shift in diatom community composition towards more heavily silicified species such as *Skeletonema* and *Pseudo-nitzschia* (Fig A1) (Martin-Jézéquel et al., 2000). The shift in community composition provides an explanation for the absence of a correlation between experimental phase and silicification for each genus. We note that, although not statistically significant, a general trend of increasing silicification soon after nutrient addition appears to be present for each genus. Similar trends have been observed in previous work,

with nitrate and phosphate limitation influencing silicification, however this is often secondary to other macro and micro nutrients such as silicate (Durkin et al., 2013; Shimada et al., 2009). Finally, we note that silicification was greatest on days 41 and 45 when nutrients were approaching depletion. Thus, it is possible that towards the end





of the experiment when nutrients were depleted, growth rates slowed significantly resulting in an increase in silicification.

**4.3 Effect of carbonate chemistry manipulations on diatom silicification**

We detected no significant relationship between total alkalinity and genus specific silicification in this mesocosm study, with the exception of *Pseudo-nitzschia* which was more heavily silicified in higher alkalinity treatments. Previous physiological studies have found tight links between diatom silicification and components of the marine carbonate chemistry system ($CO_2$ and pH) (Gao et al., 2014; Hervé et al., 2012; Li et al., 2019; Petrou et al., 2019; Zepernick et al., 2021). However current research presents some inconsistencies in the relationship between silicification and carbonate chemistry. Petrou et al. (2019) found that silicification decreased at increased $pCO_2$ and low pH expected as a result of ocean acidification. In contrast, research conducted by Li et al. (2019) and Zepernick et al. (2021) found the opposite with silicification decreasing at increasing pH/alkaline conditions. It is important to note that both Li et al. (2019) and Petrou et al. (2019) simulated ocean acidification, increasing $pCO_2$ while total alkalinity remained constant. In contrast, Hervé et al. (2012) and Zepernick et al. (2021) altered the carbonate chemistry of their respective media via additions of NaOH and HCl thereby manipulating the concentrations of carbon species but not altering total DIC values. This allows for a more direct comparison to be made with the results presented here as OAE in its unequilibrated form results in changes in carbon species concentrations without significant differences in DIC. Hervé et al. (2012) found silica incorporation rates to decrease from pH 6.4 to 8.2 and increase from 8.2 to 8.5. Visual inspection of the data presented by Hervé et al. (2012) suggests that the increase in silicification observed at elevated pH (8.5) was not statistically significant. In support of this, Zepernick et al. (2021) found no significant difference in the silicification of freshwater diatoms at pH 7.7 and 8.6, only between pH 7.7 and 9.2 was a significant decline in silicification observed. In our study, total alkalinity values corresponded to a pH range from approximately 8.0 to 8.75 (total scale). Thus, it is possible that the enhancement of alkalinity in our study and corresponding changes in carbonate chemistry were not extreme enough to result in a significant change in the silicification of the diatom community or individual genera. The pennate *Pseudo-nitzschia* was the only genus to exhibit a significant increase in silicification with increasing alkalinity. This relationship is similar to that observed by Petrou et al. (2019) who found several pennate species to exhibit a close relationship between silicification and carbonate chemistry conditions. Such a finding suggests that changes to silicification as a result of OAE is likely to be genus or species-specific supporting current knowledge surrounding the large variation in silicification between species (Martin-Jézéquel et al., 2000; Rousseau et al., 2002; Timmermans et al., 2004).

To conclude this section, we would like to highlight that other factors, which were not controlled for in this experiment, may have also contributed to the lack of a significant difference in silicification observed here. Silicification is influenced by a range of environmental parameters, such as macronutrient concentrations (primarily silica) (Claquin et al., 2002; Shimada et al., 2009), light intensity (Su et al., 2018; Taylor, 1985), and predation (Liu et al., 2016; Pondaven et al., 2007), which may have masked potential effects of total alkalinity on silicification. It is possible that OAE effects on silicification may be identifiable in other experimental settings with different boundary conditions and environmental controls. For example, our previous OAE study assessing the influence of a ~500 µmol kg$^{-1}$ alkalinity increase on coastal Tasmanian plankton communities found



significant effects of OAE on silicate dynamics, suggesting changes in diatom community silicification (Ferderer et al., 2022). These differences suggest that boundary conditions are important and that many studies assessing the effects of OAE on diatom communities will be needed to extract more robust response patterns across a range

of conditions, consistent with conclusions drawn from a synthesis on diatoms in the context of ocean acidification (Bach and Taucher, 2019).

**5 Implications for the implementation of OAE and outlook**

In conclusion, our study underlines that the use of silicate-based minerals for OAE can significantly affect silicification of the diatom community and specific genera. This result was expected and consistent with our

current understanding of silica effects on diatom communities (Baines et al., 2010; Egge and Aksnes, 1992; Hauck et al., 2013; Tréguer et al., 2021). The significant influence of $Si(OH)_4$ on diatom silicification is not surprising, as it can be a limiting nutrient for diatoms, with previous studies having shown the benefits of increased $Si(OH)_4$ concentrations as a result of olivine-based mineral dissolution (Baines et al., 2010; Hutchins et al., 2023; Martin-Jézéquel et al., 2000; Wischmeyer et al., 2003).


In contrast to the clear effects of silica fertilisation on silicification, we found no evidence to suggest that the enhancement of seawater alkalinity by $0 – 600$ µmol kg$^{-1}$ affects community level silicification, and limited evidence to suggest it influences genera-specific silicification. The lack of a clear alkalinity effect on silicification was unexpected, especially in the higher alkalinity treatments, which corresponded to a substantial change in

carbonate chemistry conditions. It is important to note that this is a relatively extreme level of OAE, yet only one genus, *Pseudo-nitzschia*, exhibited significant changes in silicification as a result of the changes in carbonate chemistry. Real world applications are predicted to employ significantly less extreme perturbations of the marine carbonate chemistry system (apart from sites of direct alkalinity addition). As such, one might hypothesize that impacts may be less significant. Furthermore, such perturbations would be relatively short lived in real world

applications since dilution of the perturbed water bodies occurs, unlike the sustained changes observed within mesocosms presented here (He and Tyka, 2023; Wang et al., 2023). Irrespective of this, there is substantial empirical evidence suggesting that changes in carbonate chemistry through ocean acidification will influence diatom communities, their growth, various aspects of silicification (e.g. rate, degree of) and subsequently silicate cycling in the ocean (Bach and Taucher, 2019; Gao et al., 2014; Li et al., 2019; Milligan et al., 2004; Petrou et al.,

2019; Taucher et al., 2022). Additionally, our previous work has shown that an increase in total alkalinity of ~500 µmol kg$^{-1}$ has a significant influence on the uptake of silica and production of BSi in a coastal phytoplankton community (Ferderer et al., 2022). The mixed outcomes observed here and in the limited OAE studies so far suggest that the responses of diatoms will differ and be dependent on the community and environmental boundary conditions. More community studies, ideally with closely aligned experimental setups, will be needed to discern

whether the response of diatoms to OAE forms any robust patterns. Such ecological observations subsequently need mechanistic underpinning, potentially achievable through the intelligent design of physiological experiments (Collins et al., 2022). Ultimately, the goal should be to provide predictive understanding of the role of diatoms and eventually all major functional plankton groups under the differing strategies of OAE.





**Appendix A**

**Table A1.** Linear mixed effects model looking at the effect of the mineral source type on silicification during nutrient replete conditions days 28 – 39.

| Source of variation | df | F-value | P-value |
|---|---|---|---|
| **Alkalinity source mineral (Ca or Si)** | 1,25 | 2.01 | 0.17 |
| **Genus** | 5,4423 | 344.05 | <0.001* |
| **Total alkalinity** | 1,25 | 0.11 | 0.74 |
| **Alkalinity source mineral*Genus** | 5,4423 | 19.09 | <0.001* |
| **Total alkalinity*Genus** | 5,4423 | 6.86 | <0.001* |

*Genus was included within the model as it explained a significant proportion of variance allowing for correct model fitting. *P<0.05*





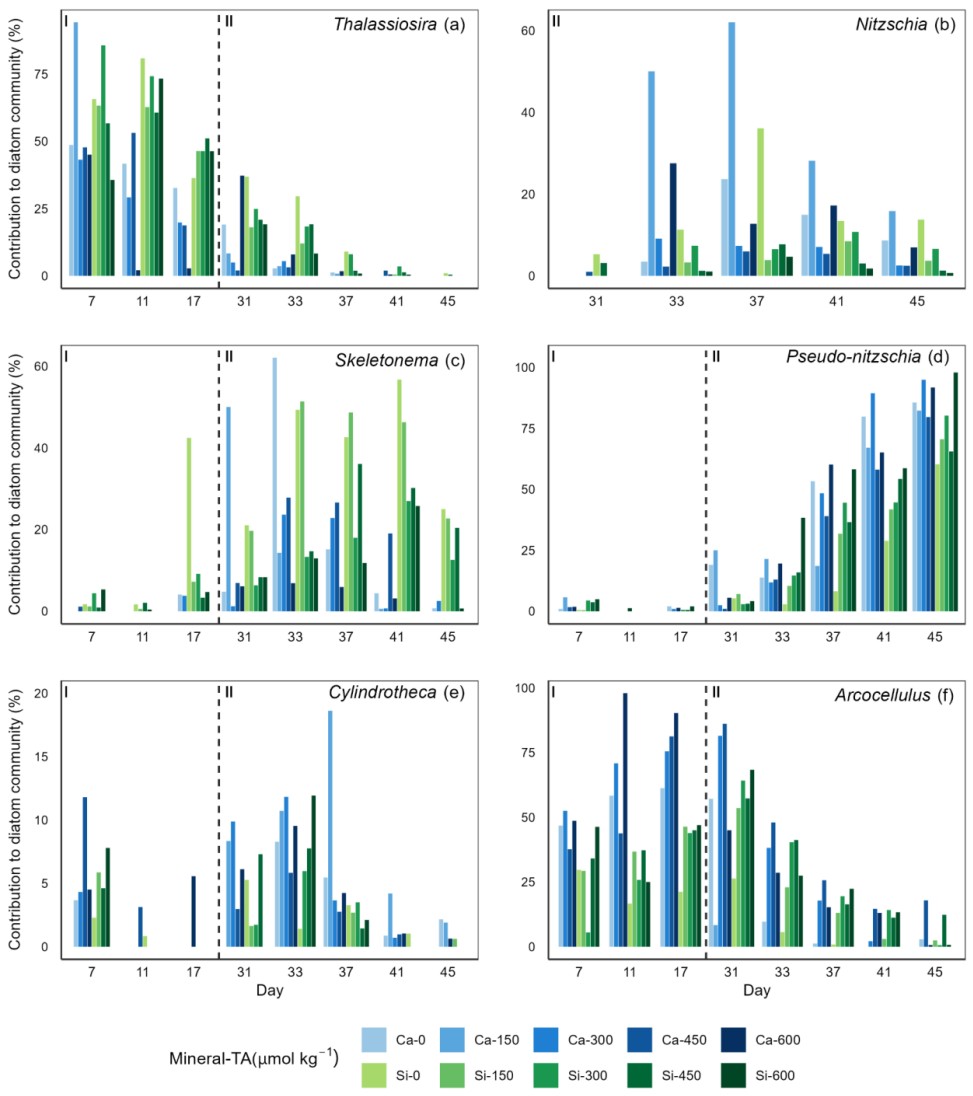


**Figure A1.** The percentage contribution of major diatom genera to the total diatom community (cell numbers) as a function of alkalinity mineral source and total alkalinity over the two experimental phases separated by the dashed vertical line (I pre-nutrient addition and II post-nutrient addition)for: **(a)** *Thalassiosira*, **(b)** *Nitzschia*, **(c)** *Skeletonema*, **(d)** *Pseudo-nitzschia*, **(e)** *Cylindrotheca*, **(f)** *Arcocellulus*.




*Data availability*

Data are available from the Institute for Marine and Antarctic Studies (IMAS) data catalogue, University of Tasmania (UTAS) https://doi.org/10.25959/G3FN-HE45.

*Author contributions*

UR designed the mesocosm experiment and AF designed the experiment assessing silicification rates with input from LTB and KGB. AF was responsible for the investigation, data curation, formal analysis, and writing. AF wrote the manuscript with contributions from LTB, KGS, UR, KGB and ZC.

*Competing interests*

The contact author has declared that none of the authors has any competing interests.

*Acknowledgements*

We would like to thank all participants of the campaign who assisted in sampling and general scientific discussion throughout. We would also like to acknowledge Juliane Tammen and Peter Fritzsche for the measurement of dissolved inorganic nutrients, Leila Kittu, Anna Groen, Lucas Krause, Jule Ploschke and Kira Lange for their

measurements of particulate matter, Jana Meyer for the measurement of sedimented material, Julieta Schneider for the measurement of carbonate chemistry, Andrea Ludwig for her organisation and management of the project and staff members at the research station for their assistance in daily tasks.

*Financial support*

This study was funded by the OceanNETS project ("Ocean-based Negative Emissions Technologies – analysing

the feasibility, risks and co-benefits of ocean-based negative emission technologies for stabilizing the climate", EU Horizon 2020 Research and Innovation Programme Grant Agreement No.: 869357), and the Helmholtz European Partnering project Ocean-CDR ("Ocean-based carbon dioxide removal strategies", Project No.: PIE-0021) with additional support from the AQUACOSM-plus project (EU H2020-INFRAIA Project No.: 871081, "AQUACOSM-plus: Network of Leading European AQUAtic MesoCOSM Facilities Connecting Rivers, Lakes,

Estuaries and Oceans in Europe and beyond"). Additional funding was supplied by a Future Fellowship (FT200100846) awarded to LTB by the Australian Research Council. This research was also conducted while Aaron Ferderer was in receipt of an Australian Government Research Training Program (RTP) scholarship.

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
