# Peer review of "Investigating the effect of silicate- and calcium-based ocean alkalinity enhancement on diatom silicification"

_Biogeosciences, 2023_

## Author Response (AR4)

Peer review round 1

Dear referees,

We thank you for your comments on our manuscript. We appreciate the time and effort you have dedicated to providing valuable feedback on our manuscript. Here are our point-by-point responses to your comments.

**Comment 1**: Lines 297-301, "Mesocosms in the Si-OAE treatment appeared to exhibit slightly higher and somewhat earlier peaks in Chlorophyll a when compared to the calcium-based treatment (Fig. 2c, phase II). This trend is supported by the uptake of NO-3 with mesocosms in the Si-OAE treatment, exhibiting marginally faster depletion of NO-3 in comparison to the Ca-OAE treatment." Qualitative statements like this are found throughout the manuscript but are not very informative. What exactly is meant by 'slightly higher', 'somewhat earlier', 'marginally faster'. These are statements that can easily be more quantitative by reporting the actual data. How much higher is chl in the Ca-OAE treatment vs Si-OAE treatment? Except for Si-150, chl concentrations look fairly similar between treatments. A crude rate of NO3 drawdown can be calculated rather than simply reporting "a marginally faster depletion of NO3".

**Response 1**: The reviewer addresses a valid point, and we agree that the addition of values to the text can improve the manuscript. As such we have adjusted the lines highlighted here to reflect the reviewers concern , (Lines 297 -301) "Chlorophyll a peaked between days 39 and 49 for all mesocosms with the Si-OAE treatment exhibiting a marginally higher mean peak in chlorophyll a (Si-OAE treatment: $3.78 \pm 1.7$ µg L$^{-1}$, Ca-OAE treatment: $2.97 \pm 0.3$ µg L$^{-1}$, Fig. 2d). A similar trend is observed in the uptake of NO$^-_3$ with mesocosms in the Si-OAE treatment, exhibiting marginally faster depletion of NO$^-_3$ (-0.192 µmol L$^{-1}$ per day) in comparison to the Ca-OAE treatment (-0.158 µmol L$^{-1}$ per day).". We have also made adjustments throughout the manuscript where appropriate e.g. Lines 294 – 296: "Chlorophyll *a* remained low until day 33, after which it increased across all mesocosms at rates between $0.03 – 0.68$ µg L$^{-1}$ per day (Fig 2d).".

**Comment 2:** Line 303-304 "However, in the Si-OAE treatments concentrations of Si(OH)4 were noticeably lower in mesocosms with high alkalinity (Fig 2c)". Again, this should be a quantitative reporting of the data that doesn't require the reader to figure out what is 'noticeable' and what is not. How much does potential dissolution of bSi that formed during the experiment contribute to the higher concentration of Si(OH)4? Peaks in water column bSi were measured around day 40 and after, which in some treatments later declined. Could this have dissolved back into the water column? With the data available, they should be able to at least provide an upper limit of how much Si(OH)4 would have increased if all of the bSi had dissolved.

**Response 2:** We appreciate the reviewers comment on the inclusion of quantitative reporting. As such we have adjusted this sentence and provided values for the difference in DSi between the upper and lower alkalinity treatments " However, in the Si-OAE treatments concentrations of Si(OH)$_4$ were lower in mesocosms with high alkalinity, with a difference of 2.45 µmol L$^{-1}$ between the $\Delta 0$ and $\Delta 600$ µmol kg$^{-1}$ alkalinity treatments (Fig 2c).". The reviewer also questions if the dissolution of BSi in the water column could be responsible for the observed differences in DSi concentrations within mesocosms. This is highly unlikely considering differences between mesocosms appeared directly after the addition of Si(OH)$_4$ on day 7 and remained until day 40 (see line

304). This trend is mirrored by concentrations of dissolved silicate with less dissolved silicate in treatments that exhibited greater concentrations of BSi during the first 37 days of the experiment. Due to the lack of differences being observed in DSi and/or BSi concentrations during the first 37 days of the experiment it is safe to suggest that the uptake of DSi was not large enough in any mesocosm to allow for any significant amount of BSi to dissolve back into the water column. Towards the conclusion of the experiment there was significantly more BSi in the water column, however as illustrated by the sediment data majority of this was removed from the water column and deposited in the sediment trap.

**Comment 3:** With respect to the PDMPO data, there are missing methodological details that make it impossible to evaluate the quality of this data. I could not find any information (in the methods, results section or figure captions) on the number of cells they imaged for each treatment and each genera. There is also no information on how they distinguished each genera which is already very hard to do by brightfield microscopy but I imagine is even harder to do by epifluorescence microscopy. What metrics were used to categorize cells into the different genera? Was there a quantitative metric (e.g. size cutoff, diameter) that was used? As even as they indicated, Pseudo-nitzschia and Nitzschia are difficult to discern, yet they provide data for each, so how were these two distinguished? These data form nearly the entirety of their conclusions yet not one single representative image was provided.

**Response 3:** We thank the reviewing for highlighting that the methods section requires more information to improve clarity. As such we have adjusted lines 236 - 237 to reflect the reviewers concern "The entirety of each filter was systematically scanned at ×200 magnification, when a cell was located, it was imaged at ×400 magnification with all cells that were appropriate for measurements (e.g., not overlapping or partially destroyed cells) included in analysis. This gave final counts for each genus ranging from 1 – 176 cells per mesocosm on any given day.". We have included a table in the supplement which has each genus and a representative image of that genus.

In relation to the query on cell identification we identified cells based on their shape, morphology, frustule characteristics (which can be more distinct when using PDMPO) and size. In addition to this daily light microscopy was conducted by other participants of the campaign which was used as a reference for the classifications of genera described here.

We agree that it is often very difficult to differentiate between *Nitzschia* and *Pseudo-nitzschia*. Our classification between the two genera was made based on the species present in daily light microscopy and the size of the cells. We agree that it is possible that not all cells in a group are indeed of that genus based on this method of classification and have made changes to reflect this. Line 237 " Cells were, when possible, identified to genus level as brightfield imaging was not possible on the black polycarbonate filters and fluorescent images did not provide enough detail for accurate identification beyond this level. However, in instances where differentiation between genera was difficult or impossible to complete with high confidence cells were classified based on significant differences in the shape, size and/or details of the frustule of cells. Each genus/group is therefore comprised of similar cells which show distinct differences in characteristics which influence the fluorescence of cells (Table S1)."

**Comment 4:** I also question the use of epifluorescence microscopy as a reliable method for quantifying 'silicification' (as defined in this study). The other studies that have used PMDPO to quantify bSi production have used confocal microcopy which allows imaging and 3D reconstruction of the entire cell. When diatoms get physiologically stressed the cell can elongate (especially centric diatoms). This would only be seen if the cell is being image from the girdle band view. Was any attempt made to image, for example, Thalassioria cells only in girdle band or valve view? As epifluroscence microscopy is not imagine in a single plane, how certain can the authors be that their measurements of silicification are truly representative?

**Response 4:** We thank the reviewer for their comment. We would like to highlight that epifluorescence microscopy has been used in several studies which have utilised PDMPO to effectively quantify single cell silicification e.g. (Znachor and Nedoma, 2008; Ichinomiya et al., 2010; Durkin et al., 2012; Saxton et al., 2012; Znachor et al., 2013, 2015; Petrou et al., 2019; Lafond et al., 2020, 2019; Zepernick et al., 2021; Iluz et al., 2009). The primary reason for using epifluorescence microscopy over confocal microscopy which provides more detailed images is that epifluorescence microscopy allows for the analysis of a large number of cells, providing a more robust estimation of silicification for a given group/taxa which was the primary aim of this study as stated in lines 251 – 252.

We agree that it is entirely possible that cell morphology changed as a function of stress/experimental treatments. However, this would not impact the results of our study as all results are corrected for cell size and given as mean fluorescence per cell (see lines 243 – 244). Furthermore, the measurement of fluorescence using epifluorescence microscopy is not dependent on cell orientation as explained by (Lafond et al., 2019) "superimposed fluorescence in the vertical plane increases the fluorescence in the x–y plane and reduces measurement biases". As such if a cell was viewed in girdle band or valve view its fluorescence would not be significantly different and any change in morphology as a result of stress and therefore the experimental treatments would be measured as a change in silicification.

**Comment 5:** I find the color scheme incredibly difficult to discern and there are so many lines or boxplots on a single graph that there is no way to look at each graph and walk away with any sort of conclusion. The data presentation needs improvement.

**Response 5:** We thank the reviewer for highlighting this point. We have adjusted all box plots in line with this comment. However, we are unable to change the colour scheme as this was decided upon by the organisers and participants of the campaign to ensure that all manuscripts utilised the same colour scheme (Blue representing the calcium-based treatment, green representing the silicate-based treatment and the specific shades/gradient of each colour for the alkalinity treatments).

**Comment 6:** Line 386 "Visual inspection of Figure 6b suggests that BSi in the sediment trap increased with alkalinity, however this was not statistically significant for any phase of the experiment (Table 3)."
What increase with alkalinity are the authors referring to? It looks to me that during Phase I and the first part of Phase II, there is more bSi in the sediment in the high alkalinity Si-OAE treatment compared the low alkalinity, but by the end of the experiment, bSi in the sediment in the low alkalinity treatment was higher. Thus, it is not

clear what exactly the authors want the reader to take away from a statement like this especially in light of the differences not being statistically significant.

**Response 6:** We thank the reviewer for this query. The increase in "BSi with alkalinity" refers to the fact that there is more BSi in the Si - $\Delta$600 µmol kg$^{-1}$ mesocosm and less in mesocosms with lower alkalinity. To clarify this, we have changed this sentence to reflect the reviewer's concern. Line 386 "During phase I there was a 0.47 µmol L$^{-1}$ difference in the amount of BSi accumulated in the sediment trap between the highest ($\Delta$600 µmol kg$^{-1}$) and lowest ($\Delta$0 µmol kg$^{-1}$) alkalinity levels in the Si-OAE treatment. However, there was less variability between the $\Delta$150, $\Delta$300 and $\Delta$450 alkalinity treatments, contributing to the non-significant effect of alkalinity on BSi accumulation in the sediment trap (Table 3). Furthermore, irrespective of experimental phase, there was no significant relationship between alkalinity and BSi accumulated in the sediment."

**Comment 7:** I'm not exactly sure what to think about all of the outliers in some of their box plot data. Why are there so many in some cases (e.g. Fig 4)? Here is where knowing the total number of cells that the box plots represent would be helpful is assessing the relevance of the outliers.

**Response 7:** We thank the reviewing for highlighting this point and understand that if the reader is not familiar with the dataset this may be confusing. In the case of Figure 4 this figure is illustrating the silicification value of all cells found on filters for a given mesocosm on a given day described as "Single cell silicification of the diatom community" in line 351. Different genera, species and even single cells of the same species often differ in the amount of silica they incorporate into their frustule. These variations often occur over several orders of magnitude, as shown in Figure 2 Petrou et al. (2019) and display significant inter- and intra-specific variability as shown in Figure 2H Ajani et al. (2021). Figure 4 illustrates changes within the entirety of the diatom community resulting in many outliers as cell fluorescence ranges from ~1.17 for weakly silicified groups such as *Cylindrotheca* to ~95.95 for more heavily silicified groups such as *Skeletonema*. In this scenario we are unsure how knowing how many cells are represented by a plot would help readers understand the presence of the outliers better. As we have grouped cells to approximately genus level it is inevitable that there will be some variation in silicification and thus outliers are to be expected in both genus specific and community plots.

**Comment 8:** Line 403-404 "The results of our study revealed silicate based OAE to significantly increase silicification in the diatom community and all genera with the exception of Cylindrotheca".
This is a bit misleading and confusing because the increase in silicification was due to an increase in the availability of dissolved silicon, not because of increased alkalinity, which the authors point out on lines 406-407. So really it's the addition of olivine that increased silicification, not the alkalinity.

**Response 8:** We appreciate the reviewer for highlighting this and have made relevant changes. "Our results revealed silicate fertilisation associated with silicate-based OAE to significantly increase silicification in the diatom community and all genera with the exception of *Cylindrotheca*.".

**Comment 9:** Lines 452-454, "As such, it is possible that the low concentrations of Si(OH)4 observed in the Ca-OAE treatment, resulted in diatoms prioritising growth over silica incorporation leading to significantly less silicification when compared to the Si-OAE treatment."
This should result in a higher growth rate in Ca-OAE treatments. Is this the case? I couldn't find any data reported about growth rates but these could be calculated from chl or bSi concentrations. In the PDMPO images, were the cells partially or fully labeled ?

**Response 9:** Unfortunately, this is largely speculation, as we did not assess growth rates of the diatom community. We are unable to use chlorophyll a data as there were other functional groups which significantly contributed to this value. As such, it would not be appropriate to use this data to estimate diatom growth rates. It is also not possible to use BSi for two reasons the first being that the amount of BSi in the Ca and Si treatments varied significantly and significantly influenced the amount of BSi in mesocosms, secondly it is possible for BSi and growth rates to decouple as shown by Paasche (1973), Brzezinski et al. (1990) and Guillard et al. (1973). As such the use of either BSi or chlorophyll a to estimate diatom growth rates would be inappropriate in this instance. However, we have added the following sentences in response to the reviewer's concerns "Detailed measurements of diatom growth (not assessed here) would be required to confirm this hypothesis. We recommend future experiments consider this and assess diatom growth alongside measures of silicification to enable the exploration of potential trade-offs between growth and silicification."

**Comment 10:** Line 483-484. Here is another reference to growth rates but I cannot find any data on growth rates in the different treatments.

**Response 10:** We thank the reviewer for highlighting this and agree that in this instance this reference to growth rates does not benefit the discussion and we have removed this sentence.

**Comment 11:** Line 500, "Visual inspection of the data presented by (Hervé et al., 2012). suggests that the increase in silicification observed at elevated pH (8.5) was not statistically significant."
Visual inspection cannot discern whether something is statistically significant or not.

**Response 11:** We agree with the reviewer and will adapt this sentence to reflect their concern. "Visual inspection of the data presented by Hervé et al. (2012) showed that the incorporation of silica into a cell at pH 8.5 was marginally less than that at lower pH values (no statistics were provided for this measurement in the cited article)."

**Comment 12:** "Silica" is a solid and should not be used when referring to the dissolved silicon (e.g. Line 517, when referring to silica as a macronutrient). This should be fixed throughout the manuscript.

**Response 12:** We agree with the reviewer and will change this to reflect their concern.

**Reviewer #2**

**Comment 1:** "...genera specific silicification also varied significantly between alkalinity sources..." and, further on, "...No other genera displayed significant changes in silicification as a result of alkalinity increase..." This seems a little contradictory on first reading; even if we understand that the main effect is mainly constrained by the type of addition: Ca-OAE versus Si-OAE; i.e. the addition of silicon stimulates silicification, which is not really a surprise?

**Response 1:** We agree with the reviewer and have adjusted the abstract to reflect their concern: " Silicification was significantly greater in the silicate-based mineral treatments, with all genera except *Cylindrotheca*, displaying an increase in silicification as a result of the increased concentration of silicate. *Pseudo-nitzschia* and *Nitzschia* were the only genera directly affected by alkalinity."

**Comment 2:** Setup of OAE treatments and nutrient fertilization- Could the authors comment on the lack of a "control" mesocosm (no addition of either Ca-OAE or Si-OAE)?

**Response 2:** We appreciate the reviewers concern and would like to highlight that there was indeed a control for both treatments (see line 121 and 126). However, we appreciate that this is not clear. As such we have adapted line 121-122 "Mesocosms were split into two treatment groups; a calcium-based (Ca-OAE) treatment (N =5) and silicate-based (Si-OAE) treatment (N =5) with one mesocosm in each group serving as a control."

**Comment 3:** 1- It would be clearer to specify in the paragraph that genus was indeed included directly - as a (random?) factor with a potential effect - in the structure of the general linear mixed-effect model (instead of building several genus-specific models) ? Notably page 13, table 1, genus is included in the general model.

**Response 3:** We thank the reviewer for this comment and agree that a single model with post hoc testing to reveal the significance of interactions at the level of factor would be a simpler method for statistical analysis. As such we will remove the genus-specific models and associated table. This will be replaced with the community-based model and post hoc testing using "emmeans" and "emtrends". Minor adjustments to the text and results tables will follow this change.

**Comment 4:** What do you mean by "several mixed models"? Is it different model structures with different a priori hypotheses (random cross-effects, nested effects,...)?

**Response 4:** This issue is resolved via changes made and outlined in response 3.

**Comment 5:** "Mesocosms in the Si-OAE treatment appeared to exhibit slightly higher and somewhat earlier peaks in Chlorophyll a... "This trend is supported by the uptake of $NO_3^-$ ...exhibiting marginally faster depletion..." Is there any way of quantifying these qualitative trends? Are they significant or not?

**Response 5:** We thank the reviewer for highlighting this issue. We did not run statistical analysis on the nutrient or Chlorophyll data as this was provided by external sources and will be published in other manuscripts at a later date. However, in line with the comments provided by reviewer 1 we have adjusted this section to reflect both reviewer's concerns.

**Comment 6:** Given the differences in silicate concentrations between +Ca and +Si media, isn't it expected that the silicification rate would be significantly higher in +Si media?

**Response 6:** Yes, it was expected.

**Comment 7:** If not addressed in the text, explain why there is no phase effect for the genus Nitzschia

**Response 7:** In line with the reviewers comment 3 this is no longer relevant to the manuscript as no interaction between genus and phase is detected within the community model and therefore no phase effect for any genus.

**Comment 8:** In Fig. 6, it might be insightful to estimate slopes during exponential growth phases after day 35 and compare them. The same could be done for Figs. 2a-2d.
Is there evidence of alkalinity affecting nutrient consumption rates or phytoplankton growth?

**Response 8:** We thank the reviewer for their comment however it is not within the scope of this manuscript to assess the consumption of nutrients or growth of the phytoplankton in this experiment. This will be part of a manuscript from our colleagues at a later date and we feel that inclusion of such data does not contribute significantly to the aim of this manuscript which was to assess the influence of mineral based OAE on taxa specific rates of silicification in the diatom community.

**Comment 9:** Page 17-18: cf. previous comment: the fact that +Si treatment stimulates the rate of silicification could be expected a priori ?

**Response 9:** Yes, we agree and did except the Si treatment to stimulate rates of silicification if Si was significantly low enough in the Ca treatment. Please see lines 80 – 84, 445 – 450, 538 – 541 for our discussion on this.

**Comment 10:** Do the authors have any hypotheses to explain the fact that Pseudo-nitzschia is more sensitive to a variation in alkalinity ?

**Response 10:** We thank the reviewer for this query. Unfortunately, we are unable to provide a hypothesis at this point in time due to the lack of physiological studies exploring the influence of elevated pH/low DIC and high alkalinity on specific phytoplankton species. However, microcosm studies (unpublished) conducted at our institute have also revealed this genus to be negatively impacted by alkalinity and we are currently conducting studies to pursue this interesting finding further.

**Comment 11:** "(2) that there was pH dependent, inorganic precipitation of amorphous silicate, with these precipitates sinking out into the sediment trap"

Was the pH measured in the mesocosms?

**Response 11:** pH was not measured in situ, but samples were taken and measured via appropriate methods on land (spectrophotometric). This data will form part of a manuscript which exclusively explores the carbonate chemistry changes within mesocosms.

**Comment 12:** "Diatoms in the Sai-OAE treatment incorporated considerably more silica over the 24-hour incubation period resulting in increased silicification. This outcome was not unexpected, especially considering the consistently low (likely limiting) concentrations of $Si(OH)_4$ observed in the Ca-OAE".

I'm not sure to understand the argument here? why was the increased silicification rate in the Si-enriched mesocosms not expected?

**Response 12:** Indeed, increases in silicification were expected as stated in line 438 " This outcome was not unexpected…". We appreciate that this wording may be confusing and have changed this in light of the reviewer's comment "This outcome was expected…". Although obvious we feel that it is important to state this significant finding as it relates directly to the selection of specific mineral types for use in OAE.

References

Ajani, P. A., Petrou, K., Larsson, M. E., Nielsen, D. A., Burke, J., and Murray, S. A.: Phenotypic trait variability as an indication of adaptive capacity in a cosmopolitan marine diatom, Environmental Microbiology, 23, 207–223, https://doi.org/10.1111/1462-2920.15294, 2021.

Brzezinski, M. A., Olson, R. J., and Chisholm, S. W.: Silicon availability and cell-cycle progression in marine diatoms, Marine Ecology Progress Series, 67, 83–96, 1990.

Durkin, C. A., Marchetti, A., Bender, S. J., Truong, T., Morales, R., Mock, T., and Virginia Armbrust, E.: Frustule-related gene transcription and the influence of diatom community composition on silica precipitation in an iron-limited environment, Limnology and Oceanography, 57, 1619–1633, https://doi.org/10.4319/LO.2012.57.6.1619, 2012.

Guillard, R. R. L., Kilham, P., and Jackson, T. A.: Kinetics of Silicon-Limited Growth in the Marine Diatom Thalassiosira Pseudonana Hasle and Heimdal (=cycloTELla Nana Hustedt)1,2, Journal of Phycology, 9, 233–237, https://doi.org/10.1111/j.1529-8817.1973.tb04086.x, 1973.

Hervé, V., Derr, J., Douady, S., Quinet, M., Moisan, L., and Lopez, P. J.: Multiparametric Analyses Reveal the pH-Dependence of Silicon Biomineralization in Diatoms, PLOS ONE, 7, e46722, https://doi.org/10.1371/JOURNAL.PONE.0046722, 2012.

Ichinomiya, M., Gomi, Y., Nakamachi, M., Ota, T., and Kobari, T.: Temporal patterns in silica deposition among siliceous plankton during the spring bloom in the Oyashio region, Deep Sea Research Part II: Topical Studies in Oceanography, 57, 1665–1670, https://doi.org/10.1016/J.DSR2.2010.03.010, 2010.

Iluz, D., Dishon, G., Capuzzo, E., Meeder, E., Astoreca, R., Montecino, V., Znachor, P., Ediger, D., and Marra, J.: Short-term variability in primary productivity during a wind-driven diatom bloom in the Gulf of Eilat (Aqaba), Aquatic Microbial Ecology, 56, 205–215, https://doi.org/10.3354/AME01321, 2009.

Lafond, A., Leblanc, K., Quéguiner, B., Moriceau, B., Leynaert, A., Cornet, V., Legras, J., Ras, J., Parenteau, M., Garcia, N., Babin, M., and Tremblay, J. É.: Late spring bloom development of pelagic diatoms in Baffin Bay, Elementa, 7, https://doi.org/10.1525/ELEMENTA.382/112521, 2019.

Lafond, A., Leblanc, K., Legras, J., Cornet, V., and Quéguiner, B.: The structure of diatom communities constrains biogeochemical properties in surface waters of the Southern Ocean (Kerguelen Plateau), Journal of Marine Systems, 212, 103458, https://doi.org/10.1016/J.JMARSYS.2020.103458, 2020.

Paasche, E.: Silicon and the ecology of marine plankton diatoms. I. Thalassiosira pseudonana (Cyclotella nana) grown in a chemostat with silicate as limiting nutrient, Marine Biology, 19, 117–126, https://doi.org/10.1007/BF00353582, 1973.

Petrou, K., Baker, K. G., Nielsen, D. A., Hancock, A. M., Schulz, K. G., and Davidson, A. T.: Acidification diminishes diatom silica production in the Southern Ocean, Nature Climate Change, 9, 781–786, https://doi.org/10.1038/S41558-019-0557-Y, 2019.

Saxton, M. A., D'souza, N. A., Bourbonniere, R. A., McKay, R. M. L., and Wilhelm, S. W.: Seasonal Si:C ratios in Lake Erie diatoms — Evidence of an active winter diatom community, Journal of Great Lakes Research, 38, 206–211, https://doi.org/10.1016/J.JGLR.2012.02.009, 2012.

Zepernick, B. N., Gann, E. R., Martin, R. M., Pound, H. L., Krausfeldt, L. E., Chaffin, J. D., and Wilhelm, S. W.: Elevated pH Conditions Associated With Microcystis spp. Blooms Decrease Viability of the Cultured Diatom Fragilaria crotonensis and Natural Diatoms in Lake Erie, Frontiers in Microbiology, 12, 2021.

Znachor, P. and Nedoma, J.: Application of the pdmpo technique in studying silica deposition in natural populations of fragilaria crotonensis (bacillariophyceae) at different depths in a eutrophic reservoir1, Journal of Phycology, 44, 518–525, https://doi.org/10.1111/J.1529-8817.2008.00470.X, 2008.

Znachor, P., Visocká, V., Nedoma, J., and Rychtecký, P.: Spatial heterogeneity of diatom silicification and growth in a eutrophic reservoir, Freshwater Biology, 58, 1889–1902, https://doi.org/10.1111/FWB.12178, 2013.

Znachor, P., Rychtecký, P., Nedoma, J., and Visocká, V.: Factors affecting growth and viability of natural diatom populations in the meso-eutrophic Římov Reservoir (Czech Republic), Hydrobiologia, 762, 253–265, https://doi.org/10.1007/S10750-015-2417-8/FIGURES/6, 2015.

Peer review round 2

Dear referees,

We thank you for your comments on our manuscript. We appreciate the time and effort you have dedicated to providing valuable feedback on our manuscript. Here are our point-by-point responses to your comments.

**General comment:** I caution the authors about making too many broad, overly-genearlized statements like in the abstract "In summary, our findings suggest limited genus-specific impacts of alkalinity on diatoms." Statements like this need to be contextualized to prevent readers from making generalized conclusions that might not be applicable in other environments. This study looked at only one aspect of diatoms, that being silicification via PDMPO incorporation. Furthermore, it was done in an environment where there were very few diatoms to begin with.

**Response to general comment:** We thank the reviewer for highlighting this and agree that such a statement is far to generalised for this specific manuscript. We have adjusted this sentence and other similar instances throughout the manuscript.

Line 35- 38: "In summary, our findings illustrate that the enhancement of alkalinity via simulated silicate- and calcium-based methods has limited genus-specific impacts on the silicification of diatoms. This research underscores the importance of understanding the full breadth of different OAE approaches, their risks, co-benefits, and potential for interactive effects.

**Comment 1:** Lines 24-33 " Silicification was significantly greater in the silicate-based mineral treatments, with all genera except Cylindrotheca, displaying an increase in silicification as a result of the increased concentration of silicate. Pseudo-nitzschia and Nitzschia were the only genera directly affected by alkalinity. The four other genera investigated here displayed no significant changes in silicification as a result of alkalinity increases between 0 and 600 µmol kg-1 above natural levels."

This is confusing. You have the first sentence saying all genera were affected except Cylindrotheca. Then it says only Pseudo-nitzschia and Nitzschia were affected. Then it says 4 genera were not affected. Please be more clear in the summary of your findings.

**Response 1:** We thank the reviewer for highlighting this and have adjusted this section in light of their comment Lines 29 – 35: "Silicification was significantly greater in the silicate-based mineral treatment, with all genera except *Cylindrotheca*, displaying an increase in silicification as a result of the increased concentration of dissolved silicate. In contrast to the effect of differences in dissolved silicate concentrations between the two mineral treatments, increases in alkalinity only influenced the silicification of two genera, *Pseudo-nitzschia* and *Nitzschia*. The four other genera investigated here (*Arcocellulus, Cylindrotheca, Skeletonema,* and *Thalassiosirra)* displayed no significant changes in silicification as a result of alkalinity increases between 0 and 600 µmol kg$^{-1}$ above natural levels.

**Comment 2:** Figures should be introduced in the order they appear in the text. (e.g. Results start with Fig. 2d as opposed to 2a).

**Response 2:** We thank the reviewer for highlighting this mistake and have adjusted the order which the figures are introduced within the text.

Line 291-300: "Concentrations of $NO_3^-$ were below detection limit, thereby constraining phytoplankton growth during phase I of the experiment (mean $NO_3^-$ day 7 – 25  = 0.004 ± 0.035 µmol $L^{-1}$) (Fig 2a). In contrast there was residual $PO_4^{3-}$ (0.021 ± 0.022 µmol$^{-1}$) and $Si(OH)_4$ (Ca-OAE treatment = 0.202 ± 0.99 µmol$^{-1}$, Si-OAE treatment = 67.929 ± 1.04 µmol$^{-1}$) which likely supported the phytoplankton community in utilising remineralised nitrogen until the addition of nutrients on day 26 and 28 (Fig 2b, 2c). Although ~75 µmol $L^{-1}$ of $Na_2SiO_3$ was added to the Si-OAE treatment, there was no discernible depletion of $Si(OH)_4$ during phase I (Fig 2c). The addition of macronutrients ($NO_3^-$, $PO_4^{3-}$, and $Si(OH)_4$) can be seen on day 26, with a secondary addition on day 28 to correct for unwanted differences in the stoichiometry between mesocosms (Fig 2). Chlorophyll *a* concentrations were relatively low at the beginning of the experiment with 1.01 ± 0.17 µg $L^{-1}$ (mean ± SD) on day 3 (Fig 2d)."

**Comment 3:** Line 299-308 starting with "Chlorophyll a peaked between days 39 and 49 for all mesocosms with the Si-OAE treatment exhibiting a marginally higher mean peak in chlorophyll a (Si-OAE treatment: 3.78 ± 1.7 µg L-1, Ca-OAE treatment: 2.97 ± 0.3 µg L-1, Fig. 2d). From just the error alone, it doesn't not look like this difference is statistically significant so therefore saying that it is marginally higher is misleading. Either provide statistical support or remove such phrases. Same comment for subsequent statements about nitrate and silicic acid.

**Response 3:** We understand the reviewers concern and have removed the sections where comparisons between treatments with no statistical analysis were made. We have kept the discussion surrounding differences in silicic acid concentrations, with a modification so that only the highest and lowest alkalinity treatments are discussed. We feel that this is important to include as it relates directly to the observed differences in starting concentrations of dissolved silicate and BSi in the sediments.

Line 304 – 309: "There was no discernible relationship between total alkalinity and chlorophyll a, $NO_3^-$ or $PO_4^{3-}$ observed across the extent of the experimental period or in a particular phase (Fig 2). However, in the Si-OAE treatments initial concentrations of $Si(OH)_4$ were lowest in the high alkalinity mesocosm, with a difference of 2.45 µmol $L^{-1}$ between the Δ0 and Δ600 µmol $kg^{-1}$ alkalinity treatments (Fig 2c). This trend appeared directly after the addition of the treatments but disappeared once nutrient uptake began (Fig 2c)."

**Comment 4:** Can the same symbols in Fig. 3 be used in Fig 2? This would improve the ability to differentiate the numerous lines shown in each graph which is currently very difficult to do. For example, it is stated that in Fig. 2c the silicic acid concentration in the higher alkalinity treatments is lower. Zooming it, it looks like one of the high alkalinity treatments has silicic acid concentrations similar to the low alkalinity, but it is difficult to match the exact color to the treatment.

**Response 4:** Yes, we agree with the reviewer and have adjusted figures 2 and 6 so that all graphs which utilise symbols to distinguish between alkalinity levels use the same symbols as figure 3.

**Comment 5:** Line 319, "small distances are observed between Pseudo-nitzschia and Nitzschia, likely due to their morphological similarities and therefore similar silica content"

Just because they have similar morphologies does not mean they have similar silica content. Furthermore, different Pseudo-nitzschia species can have different silica quotas in addition to the ability of diatoms to alter cellular silica quota.

**Response 5:** We agree that this statement is not well supported and would require further investigation. As such we have adjusted this sentence:

Line 328 – 330: *"Nonmetric multidimensional scaling (NMDS) (Fig. 3) revealed distinct distances among treatments, including different alkalinity source minerals and total alkalinity, in relation to silicification of the various diatom genera."*

**Comment 6:** The inclusion of final counts for each genus provided in the methods is appreciated (1-176 cells per mesocosm per day), but all this says is that with respect to Figs. 4 and 5, these data could represent 1 cell or 176 cells. If the authors want to make the claim that Pseudo-nitzschia and Nitzschia have significantly higher silicification with OAE, knowing the number of cells this is based on would make the conclusion more robust.

**Response 6:** We agree with the reviewer that counts for these two genera specifically would be beneficial and have now included them within the text.

Line 362 – 364: "Exploration of this interaction revealed the silicification of cells in the genus *Pseudo-nitzschia* ($N = 3510$) to be significantly influenced by alkalinity in both the Ca and Si-based treatments, with silicification increasing with increasing alkalinity (Table 3). In contrast, the genus *Nitzschia* ($N = 677$) displayed…"

**Comment 7:** Do the authors have any thoughts as to how Pseudo-nitzschia was able to increase silicification in Ca treatment where there was very little Si?

**Response 7:** We are currently investigating this difference but believe that the increase in alkalinity and subsequent decrease in $CO_2$ may have slowed growth and therefore increased the time and uptake of DSi resulting in the observed differences in silicification.

**Comment 8:** Line 419, "Our results revealed silicate fertilisation associated with silicate-based OAE to significantly increase silicification in the diatom community and all genera with the exception of Cylindrotheca."

This should be qualified by stating that this is relevant when dissolved silicon in the initial conditions are low as was the case in this study. If initial Si concentrations are high, it is possible the increase in silicification would not be seen in response to Si-OAE.

**Response 8:** We thank the reviewer for their comment and agree that if initial concentrations of dissolved silicate were high, further increases in dissolved silicate as a result of Si-OAE would most likely not increase rates of silicification. We have adjusted this section accordingly.

Lines 431 – 435: "It is important to note that low initial concentrations of dissolved silicate likely facilitated the observed increase in silicification. Under conditions where dissolved silicate is already replete, further increases due to silicate-based OAE may not result in similar increases in additional silicification. This increase in silicification was primarily a result of the difference in the dissolved silicate concentrations ($\Delta 75$ µmol kg$^{-1}$) between the silicate and calcium-based OAE treatments rather than an increase in alkalinity."

**Reviewer #2**

**Comment 1:** In the text, check nitrate and phosphate: sometimes written NO-3 and PO-34?

**Response 1:** We thank the reviewing for highlighting this and will rectify this mistake throughout the manuscript.

**Comment 2:** L. 301-303: "A similar trend is observed in NO3- uptake with mesocosms in the Si-OAE treatment, showing marginally faster NO-3 depletion (-0.192 µmol L-1 per day) compared with the Ca-OAE treatment (-0.158 µmol L-1 per day)"
Is there an error (SE) associated with these consumption rates?

**Response 2:** This section has been removed from the manuscript in line with recommendations made by reviewer 1.